# ENABLING TRUE GLOBAL PERCEPTION IN STATE SPACE MODELS FOR VISUAL TASKS

**Jie Hui**[1]     **Zhenxiang Zhang**[1]     **Wenyu Mi**[1]     **Jianji Wang**[1]*

[1] State Key Laboratory of Human-Machine Hybrid Augmented Intelligence,
  Institute of Artificial Intelligence and Robotics, Xi'an Jiaotong University, China
`huijie_@stu.xjtu.edu.cn, zzx2024@stu.xjtu.edu.cn`
`wenyu_mi@163.com, wangjianji@mail.xjtu.edu.cn`

## ABSTRACT

Despite the importance of global contextual modeling in visual tasks, a rigorous mathematical definition remains absent, and the concept is still largely described in heuristic or empirical terms. Existing methods either rely on computationally expensive attention mechanisms or are constrained by the recursive modeling nature of State Space Models (SSMs), making it challenging to achieve both efficiency and true global perception. To address this, we first propose a mathematical definition of global modeling for visual images, providing a theoretical foundation for designing globally-aware and interpretable models. Based on in-depth analysis of SSMs and frequency-domain modeling principles, we construct a complete theoretical framework that overcomes the limitations imposed by SSMs' recursive modeling mechanism from a frequency perspective, thereby adapting SSMs for global perception in image modeling. Guided by this framework, we design the Global-aware SSM (GSSM) module and formally prove that it satisfies definitional requirements of global image modeling. GSSM leverages a Discrete Fourier Transform (DFT)-based modulation mechanism, providing precise front-end control over the SSM's modeling behavior, and enabling efficient global image modeling with linear-logarithmic complexity. Building upon GSSM, we develop GMamba, a plug-and-play module that can be seamlessly integrated at any stage of Convolutional Neural Networks (CNNs). Extensive experiments across multiple tasks, including object detection, semantic segmentation, and instance segmentation, across diverse model architectures, demonstrate that GMamba consistently outperforms existing global modeling modules, validating both the effectiveness of our theoretical framework and the rigor of proposed definition. Code is available at https://github.com/Xinmu-Tantai/GMamba-GSSM

## 1 INTRODUCTION

In visual tasks, capturing long-range dependencies is essential for accurate object recognition and semantic understanding. However, despite the central role of global image modeling in improving visual perception, a rigorous mathematical definition that guarantees true global awareness is still lacking. Thus, existing studies typically rely on extensive ablation experiments, performance comparisons, or posterior feature visualizations to assess global modeling capabilities, treating global image modeling as an empirical and somewhat ambiguous concept, without systematic theoretical derivations or proofs. This lack of formal definition limits interpretability and theoretical support.

Through extensive research, existing global modeling methods can be categorized into two types: Transformers (Han et al., 2021; Liu et al., 2021) based on self-attention and Mamba (Gu & Dao, 2023) based on state space models. Transformers can capture global information through parallel modeling, but their quadratic complexity limits their applicability in resource-constrained environments (Guo et al., 2020), as shown in Fig. 1(a). In contrast, Mamba, recently proposed in Natural Language Processing (Fanni et al., 2023), offers linear complexity and efficient long-sequence modeling via a novel SSM-based architecture. The core of Mamba lies in combining SSM with a selection mechanism to selectively process input sequences in a hardware-aware manner, while utilizing

---

*Corresponding Author: Jianji Wang (wangjianji@mail.xjtu.edu.cn).

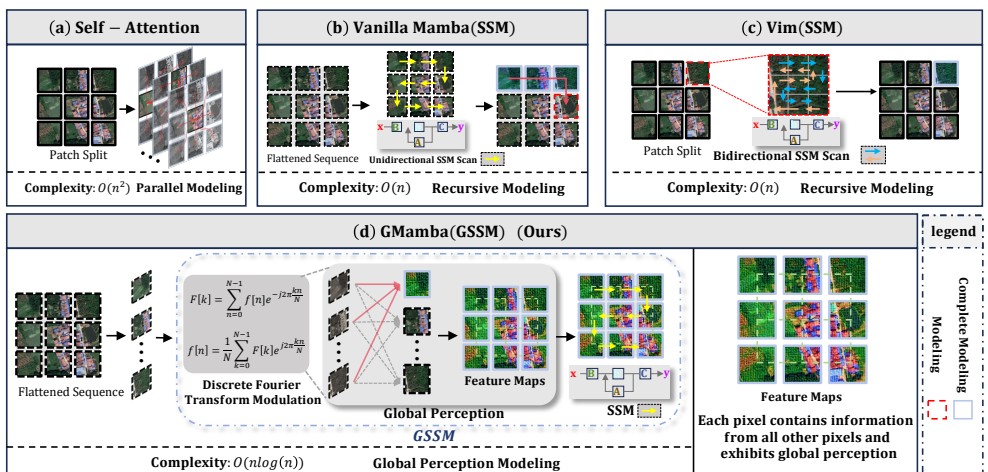

Figure 1: Existing global modeling approaches include: (a) Self-Attention, capturing global dependencies in parallel with $O(n^2)$ complexity; (b) Vanilla Mamba, adopting recursive SSMs with linear $O(n)$ complexity but limited receptive fields; (c) Vim, which enhances modeling capacity via Bidirectional SSMs; and (d) our proposed GMamba, introducing DFT-based pre-modulation to embed global perception into features, guiding SSMs for truly global modeling.

SSM's recurrence to model image information, as shown in Fig. 1(b). Recently, many studies (Wang et al., 2025a; Xu et al., 2024; He et al., 2025) have applied Mamba to vision tasks, fully leveraging its linear complexity advantage. However, applying Mamba to vision tasks faces two critical issues: (1) The modeling process of SSM employs a step-by-step state update mechanism, where each step's computation only depends on the current input and historical state, meaning the model must rely on multi-step indirect propagation to obtain global context. Its perceptual capability remains limited to local rather than explicit global modeling. (2) SSM's state update mechanism is suitable for time-series data with clear causal structure, while pixel relationships in images are non-causal, creating a structural conflict with SSM's modeling assumptions, thereby limiting its performance in image modeling. To address these issues, some methods (Li et al., 2025a; Wang et al., 2025b; Zhang et al., 2025; Pei et al., 2025; Shi et al., 2024) attempt to compensate for the limitations of unidirectional scanning in image tasks by modifying various scanning strategies. For example, Vim (Zhu et al., 2024a), as shown in Fig. 1(c). In fact, existing methods that modify various scanning strategies (Zhu et al., 2024b) have not significantly alleviated the inherent limitations of SSM.

To tackle these issues, we propose for the first time a mathematical definition of global image modeling, based on in-depth analysis of its behavior. Motivated by the extensive use of frequency-domain analysis in computer vision—particularly the DFT (Sundararajan, 2001; Pei & Yeh, 1998; Winograd, 1978), where each frequency component inherently aggregates global information and provides natural global receptive capability—we explore a frequency-domain perspective for more effective global modeling. Building on a thorough analysis of SSM mechanisms and frequency-domain transformation principles, we establish the theoretical foundation for frequency-domain modulated SSM and rigorously prove the global properties of the DFT, thereby constructing a complete theoretical framework for frequency-guided SSM. Guided by this theoretical basis, we design the GSSM module (Figs. 1(d) and 2) and demonstrate that it strictly satisfies the proposed mathematical definition of global image modeling. The module injects global semantic information into the SSM through a learnable 2D-DFT-based frequency-domain pre-modulation mechanism. By introducing global semantics at the front end and adaptively controlling the guidance and fusion between frequency-domain and spatial features, GSSM enables precise regulation of state selection and hidden-state updates in the SSM. This design allows the modeling process to go beyond historical states and, without complex scanning strategies, achieve efficient global modeling guided by global perception, while maintaining linear–logarithmic computational complexity.

Based on GSSM, we further design the GMamba block, which can be flexibly embedded into any stage of CNNs to enhance the network's understanding of global context. We conduct evaluations on multiple mainstream CNN backbones, covering four remote sensing datasets (for semantic segmentation) and MS-COCO (Lin et al., 2014) (for object detection and instance segmentation). Experimental results demonstrate that GMamba exhibits superior generalization capabilities across

different tasks and network architectures, achieving better performance than other global modeling modules while introducing fewer parameters. This validates the universality and effectiveness of the proposed theoretical framework and the rationality of our mathematical definition for global modeling in images, laying the foundation for designing globally-aware image models with stronger theoretical grounding and interpretability.

## 2 THE PROPOSED METHOD

In this section, we first propose a mathematical definition of global image modeling, then establish a complete theoretical framework for frequency-domain modulation that endows SSMs with global properties, and finally provide detailed descriptions of the proposed GMamba and GSSM modules.

### 2.1 GLOBAL PERCEPTION MECHANISM

**Global Perception Property for Images:** Here, we propose a formal definition to precisely characterize the notion of global perception in image modeling.

Motivated by the inherent unstructured nature of image data (Yu & Wang, 2025) and the fact that derivative captures the sensitivity of outputs with respect to input variables (Bishop, 2006), we introduce the following mathematical definition.

*Definition:* For an image modeling function $f : \mathbb{R}^{H \times W \times C} \to \mathbb{R}^{H \times W \times C}$ which is differentiable almost everywhere in $\mathcal{D}_f$, and a global influence function $\mathcal{I} : \{1, \ldots, H\} \times \{1, \ldots, W\} \times \{1, \ldots, C\} \to \mathbb{R}^+$ whose derivative exists at all points in $\mathcal{D}_f$, if the Frobenius norm of the derivative of $f$ with respect to any input pixel is greater than $\mathcal{I}$, then $f$ exhibits global gradient dependency.

$$\left\| \frac{\partial f(\mathbf{X})}{\partial X_{i,j,c}} \right\|_F \geq \mathcal{I}(i, j, c) > 0, \quad \forall (i, j, c) \in \mathcal{D}_f; \quad \inf_{(i,j,c)} \mathcal{I}(i, j, c) \geq \tau > 0, \tag{1}$$

where $X_{i,j,c}$ represents the pixel value at position $(i, j)$ and channel $c$ in the input image $\mathbf{X} \in \mathbb{R}^{H \times W \times C}$, $\| \cdot \|_F$ denotes the Frobenius norm of the gradient tensor which measures the degree of influence of input pixels at different positions on the output image, $\mathcal{D}_f$ denotes the set of points where both $f$ and $\mathcal{I}$ are differentiable, inf represents the infimum, and $\tau$ ensures that each input pixel exerts a stable and non-negligible influence on the reconstructed output.

*Constraint:* For function $f$, due to the absence of inherent sequential regularity in image features, $f$ should not impose strict order-dependent constraints on the input.

Our proposed mathematical definition of image global modeling, together with its constraints, establishes a rigorous and guaranteeable architectural criterion. This definition provides a fundamental framework for encompassing and evaluating existing heuristic methods. Self-attention–based approaches achieve image globality only when their learned weights happen to satisfy $\tau > 0$, yet their architectures do not enforce this property. In contrast, SSM-based models face an inherent structural conflict: their causal architectural assumption is incompatible with the non-causal nature of images, making it impossible to satisfy the non-sequential constraint while simultaneously guaranteeing a gradient lower bound (i.e., avoiding exponential decay). Thus, our definition transforms image globality from an unstable, empirically observed emergent property into a theoretical property that can be rigorously analyzed and guaranteed at the architectural design stage. This shift greatly enhances the interpretability and theoretical grounding of image global modeling and lays the foundation for designing architectures that truly provide global guarantees.

### 2.2 WHY DOES DFT ENDOW SSMs WITH TRUE GLOBAL PERCEPTION?

We conduct an in-depth analysis of the modeling mechanisms and frequency-domain modeling behavior of SSMs, thereby establishing the theoretical foundation for frequency-domain modulated SSMs. We provide two proofs: (i) 2D-DFT possesses inherent global properties in image tasks, and (ii) DFT-based modulation mechanisms can endow SSMs with true global perception capabilities.

#### 2.2.1 SSM IN FREQUENCY DOMAIN

**The modeling mechanism of SSMs.** Classical SSMs describe continuous-time systems. However, in deep learning, SSMs must be discretized by introducing a time scale parameter $\Delta \in \mathbb{R}$ and applying the widely used zero-order hold method. The discrete-time SSM is defined as follows:

$$\bar{A}(\Delta) = e^{A\Delta}, \quad \bar{B}(\Delta) = \left( \int_0^\Delta e^{A\tau} d\tau \right) B = A^{-1}(e^{A\Delta} - I)B, \tag{2}$$

$$x_t = \bar{A}(\Delta) x_{t-1} + \bar{B}(\Delta) u_t, \quad y_t = C x_t.$$

where $A \in \mathbb{R}^{N \times N}$, $B \in \mathbb{R}^{N \times M}$, and $C \in \mathbb{R}^{P \times N}$ are the continuous-time system parameters. The matrices $\bar{A}(\Delta)$ and $\bar{B}(\Delta)$ are the discretized state transition and input matrices, respectively. This discrete formulation enables efficient sequence modeling in neural networks by evolving the hidden state $x_t$ over discrete time steps with step size $\Delta$.

Meanwhile, SSM can be equivalently expressed in convolution form as:

$$y_t = \sum_{k=0}^{t} K_k u_{t-k}, \quad \text{where } K_k = C\bar{A}(\Delta)^k \bar{B}(\Delta). \tag{3}$$

**Local Perception Limitation of SSMs in Image Tasks:** SSM exhibits local perception limitations in image tasks due to its sequential processing mechanism and exponential decay of long-range influence coefficients. In image tasks, SSM flattens the pixel grid into a sequence, where the spatial distance between two pixels $(i_1, j_1)$ and $(i_2, j_2)$ in the flattened sequence is defined as $d = |i_2 \cdot W + j_2 - i_1 \cdot W - j_1|$. The influence coefficient follows the bound $|K_d| \leq |C||\bar{A}|^d|\bar{B}|$, where $\rho(\bar{A}) < 1$. As the spatial distance $d$ increases, $|K_d|$ exhibits exponential decay. Furthermore, SSM relies on sequential state transitions to capture global contextual information. While it can achieve global information access, its perception is inherently local rather than truly global modeling.

**Equivalence between SSM and Frequency Domain Representation.** We observe that SSM essentially models the input sequence through step-by-step dynamic convolution (Eq. 3), where convolution fundamentally serves as a form of filtering. From this perspective, SSM can be regarded as performing dynamic filtering on the input. Frequency-domain analysis provides an effective tool to characterize filter behavior in modeling. Based on this, we further aim to regulate the SSM modeling mechanism from a frequency-domain perspective. For a causal and stable discrete-time SSM with $\rho(\bar{A}) < 1$, we derive the frequency-domain transfer function as follows. The detailed derivation is provided in the Appendix.

$$H(\omega) = C\big(e^{j\omega}I - \bar{A}\big)^{-1}\bar{B}. \tag{4}$$

The frequency-domain transfer function of an SSM is mathematically equivalent to the discrete-time Fourier transform of its convolution kernel.

$$\hat{K}(\omega) = \sum_{k=0}^{\infty} K_k e^{-j\omega k} = C\big(I - \bar{A}e^{-j\omega}\big)^{-1}\bar{B} = e^{j\omega}C\big(e^{j\omega}I - \bar{A}\big)^{-1}\bar{B} = e^{j\omega}H(\omega). \tag{5}$$

The convergence of the geometric series $\sum_{k=0}^{\infty}(\bar{A}e^{-j\omega})^k = (I - \bar{A}e^{-j\omega})^{-1}$ is guaranteed when $\rho(\bar{A}) < 1$.

**Theoretical Foundation of Frequency-Domain Modulated SSM.** Based on the established equivalence (Eq. 5), we further present three theoretical propositions that elucidate the rationality of frequency-domain modulation in SSM.

- Information Preservation: The encoded hidden state information $x_t$ in SSMs can be transformed into a convolutional form $K_k$, where frequency-transformed information of $K_k$ directly maps with original information, ensuring frequency domain operations preserve the representational capacity of the original model.
- Bidirectional Convertibility: The dynamic convolution kernel $K_k$ of SSM and its frequency domain transfer function $H(\omega)$ form a Fourier transform pair, enabling lossless bidirectional conversion between time-domain and frequency-domain representations.
- Linear Information Propagation: Both the discrete convolution $y_t = \sum_{k=0}^{t} K_k u_{t-k}$ and the frequency-domain multiplication $\hat{Y}(\omega) = H(\omega) \cdot \hat{U}(\omega)$ are linear relations, ensuring consistent and stable linear information propagation through the network in either domain.

This theoretical foundation justifies the use of frequency-domain modulation in SSMs while preserving the original computational paradigm of SSM.

**The Global Property of the Discrete Fourier Transform:** In frequency-domain analysis, the DFT provides a natural mechanism for global modeling, as each frequency coefficient depends on all positions of the input sequence. For a one-dimensional signal $x \in \mathbb{R}^N$, the DFT is defined as:

$$\hat{X}(\omega_k) = \sum_{n=0}^{N-1} x_n e^{-j\omega_k n}, \quad \omega_k = \frac{2\pi}{N}k, \quad k = 0, 1, \ldots, N-1. \tag{6}$$

Global Property: Each frequency component $\hat{X}(\omega_k)$ depends on all positions $n$ simultaneously through the exponential terms $e^{-j\omega_k n}$. This creates an explicit global dependency structure where: $\frac{\partial \hat{X}(\omega_k)}{\partial x_n} = e^{-j\omega_k n} \neq 0$ for all frequency indices $k$ and positions $n$. Position Invariance Property: The DFT exhibits uniform positional treatment where the contribution of each position $n$ to the frequency domain representation maintains consistent magnitude $|e^{-j\omega_k n}| = 1$ across all positions. This ensures that no position is inherently privileged or degraded in the frequency domain representation. Global Reconstruction: Through the inverse DFT, any position $p$ in the reconstructed signal depends on all frequency components:

$$x_p = \frac{1}{N} \sum_{k=0}^{N-1} \hat{X}(\omega_k) e^{j\omega_k p}. \tag{7}$$

The DFT models information from all time steps and, through its position invariance property, ensures that the information at each position contributes meaningfully to the output, without imposing any restrictions on the input. Furthermore, we observe that the definition of the DFT is analogous to that of the non-local operation (Wang et al., 2018): *a non-local operation computes the response at a position as a weighted sum of the features at all positions in the input feature maps.* This further supports the global property of the DFT. In the appendix, we provide a proof of the global properties of the DFT and prove that its extension to 2D-DFT satisfies the proposed *Definition*.

**Extension to 2D-DFT for Image Tasks:** For image modeling tasks, the 2D-DFT preserves the global characteristics of the 1D-DFT while extending them to capture spatial dependencies in both dimensions (height and width). The 2D-DFT for an image $\mathbf{X} \in \mathbb{R}^{H \times W}$ is defined as:

$$\hat{\mathbf{X}}_{u,v} = \mathcal{F}_{2D}[\mathbf{X}] = \sum_{i=0}^{H-1} \sum_{j=0}^{W-1} X_{i,j} \, e^{-2\pi i \left( \frac{ui}{H} + \frac{vj}{W} \right)}, \quad u = 0, \ldots, H-1, \, v = 0, \ldots, W-1 \tag{8}$$

### 2.2.2 THEORETICAL DESIGN OF GSSM MODULE

**Global Perception of GSSM:** Based on the feasibility of frequency-domain modulated SSM theory in Sec. 2.2.1 and the global properties of the 2D-DFT as proved in Sec. 2.1, we theoretically designed the GSSM module, adopting a **frequency-guided modulation strategy** to precisely control the update process of input states and hidden states in SSM. The following is our theoretical design of GSSM. First, we apply the 2D-DFT to the input features $X$ to inject global awareness.

$$\mathbf{F}_{freq} = \mathcal{F}_{2D}^{-1} \left[ \mathcal{H}_{freq} \left( \hat{\mathbf{X}} \right) \right], \quad \hat{\mathbf{X}} = \mathcal{F}_{2D}(X) \tag{9}$$

where $\mathcal{F}_{2D}$ and $\mathcal{F}_{2D}^{-1}$ denote the 2D-DFT and its inverse transform, and $\mathcal{H}_{freq}(\cdot)$ represents learnable frequency domain transformations.

Then we employ an adaptive modulation mechanism to adjust contributions of original features and frequency-domain global features across different inputs and network stages.

$$\mathbf{X}_{modulated} = \mathcal{G}(X, \mathbf{F}_{global}) = \alpha_1(\mathbf{F}_{global}) \odot X + \alpha_2(\mathbf{F}_{global}) \odot \mathbf{F}_{global} \tag{10}$$

where $\mathbf{F}_{global} = \mathrm{Concat}[X, \mathbf{F}_{freq}]$ aggregates the original features with frequency-derived global features, and $\alpha_1, \alpha_2 \in (0, 1)$ are adaptive modulation weights derived from $\mathbf{F}_{global}$. This adaptive learning design can be effectively applied to different inputs in visual image tasks.

The complete GSSM process can be expressed as:

$$\mathbf{Y} = \mathrm{SSM}(\mathcal{G}(X, \mathcal{F}_{2D}^{-1}[\mathcal{H}_{freq}(\mathcal{F}_{2D}[X])])) \tag{11}$$

where $\mathcal{G}(\cdot)$ denotes the global modulation function defined in Eq. 10, which precisely regulates input and state update processes of SSM. Although this design is simple, the effectiveness of GSSM in global modeling will be validated by both theoretical proof and experimental validation in the following subsections.

**Further justification for our theory-driven design.** We further analyze and prove that GSSM satisfies the proposed *Definition*.

*Proving for Definition:* The gradient of the GSSM output $Y_{p,q}$ with respect to the input $X_{i,j,c}$ can be decomposed, according to the chain rule, into two components: the SSM processing of the mod-

ulated input, and the global frequency-domain modulation.

$$\frac{\partial Y_{p,q}}{\partial X_{i,j,c}} = \underbrace{\frac{\partial Y_{p,q}}{\partial \mathbf{X}_{modulated}}}_{\text{SSM process}} \cdot \underbrace{\frac{\partial \mathbf{X}_{modulated}}{\partial X_{i,j,c}}}_{\text{Global Modulation}} \tag{12}$$

We first demonstrate that the **Global Modulation** satisfies *Definition*. According to the modulation formulation in Eq. 10, the gradient can be decomposed as follows:

$$\frac{\partial \mathbf{X}_{modulated}}{\partial X_{i,j,c}} = \alpha_1 \underbrace{\frac{\partial X}{\partial X_{i,j,c}}}_{\text{direct path}} + \alpha_2 \underbrace{\frac{\partial \mathbf{F}_{freq}}{\partial X_{i,j,c}}}_{\text{global frequency path}} \tag{13}$$

**Spatially Uniform Global Influence**:The gradient of $\mathbf{F}_{freq}$ with respect to $X_{i,j,c}$ can be written as:

$$\frac{\partial \mathbf{F}_{freq}}{\partial X_{i,j,c}} = \mathcal{F}_{2D}^{-1}\left[\frac{\partial \mathcal{H}_{freq}}{\partial \hat{\mathbf{X}}} \cdot \frac{\partial \hat{\mathbf{X}}}{\partial X_{i,j,c}}\right] \tag{14}$$

Since $\frac{\partial \hat{\mathbf{X}}}{\partial X_{i,j,c}}(\omega_1, \omega_2) = e^{-j(\omega_1 i + \omega_2 j)}$ and $|e^{-j(\omega_1 i + \omega_2 j)}| = 1$ for all spatial positions, the magnitude of frequency-domain coupling is location-independent. Hence we can establish a spatially uniform lower bound:

$$\left\|\frac{\partial \mathbf{X}_{modulated}}{\partial X_{i,j,c}}\right\|_F \geq \min(\alpha_1, \alpha_2) \cdot \tau > 0 \tag{15}$$

where $\tau$ is independent of spatial location $(i, j, c)$, yielding the global influence function $\mathcal{I}(i, j, c) = \min(\alpha_1, \alpha_2) \cdot \tau$.

We then demonstrate that the **SSM process** of the modulated input satisfies *Definition*. The sequential state update mechanism of the SSM models global information through an autoregressive formulation $[x_t = \bar{A}(\Delta)x_{t-1} + \bar{B}(\Delta)u_t]$. Each state $x_t$ accumulates information from all previous inputs $u_1, u_2, \ldots, u_t$ through the recurrent updates. This can be explicitly written as:

$$x_t = \sum_{k=1}^{t} \bar{A}(\Delta)^{t-k}\bar{B}(\Delta)u_k \tag{16}$$

Since the output $Y_{p,q} = Cx_T$ is calculated from the final state $x_T$ (where $T$ is the sequence length), and $x_T$ incorporates information from all input positions, we have:

$$\frac{\partial Y_{p,q}}{\partial X_{i,j,c}} = C\,\bar{A}(\Delta)^{T-k}\bar{B}(\Delta)\,\frac{\partial u_k}{\partial X_{i,j,c}} \neq 0, \tag{17}$$

where $k$ corresponds to the sequential index of pixel $(i, j, c)$ in the flattened input sequence. Moreover, each pixel in $\mathbf{X}_{modulated}$ contains information from all other pixels, enabling the SSM update process to be guided by global context, thus eliminating the constraints of recursive modeling.

Therefore, the SSM's ability to propagate information through its hidden states ensures that every input pixel contributes to the final output, establishing:

$$\left\|\frac{\partial Y_{p,q}}{\partial \mathbf{X}_{modulated}}\right\|_F \geq \beta > 0 \tag{18}$$

where $\beta$ is a positive constant determined by the SSM system matrices.

From this, we can prove that the Frobenius norm of $\frac{\partial Y_{p,q}}{\partial X_{i,j,c}}$ is greater than zero. There exists a global influence function $\mathcal{I}_1(i, j, c) = \min[\min(\alpha_1, \alpha_2) \cdot \tau, \quad \beta] > 0$ that satisfies the conditions of our proposed *Definition*.

*Analysis of Constraint:* The GSSM employs a DFT-based modulation mechanism to perform parallel global modeling of the input and further modulates the input to the SSM. As a result, the GSSM no longer relies on local state information for stepwise modeling but instead regulates the input and state update process with global awareness. Consequently, the GSSM imposes no sequential constraints and satisfies the proposed *Constraint*.

*Conclusion:* The integration of DFT-based global coupling and adaptive modulation ensures that GSSM satisfies *Definition* and *Constraint*, thereby theoretically possessing global perception.

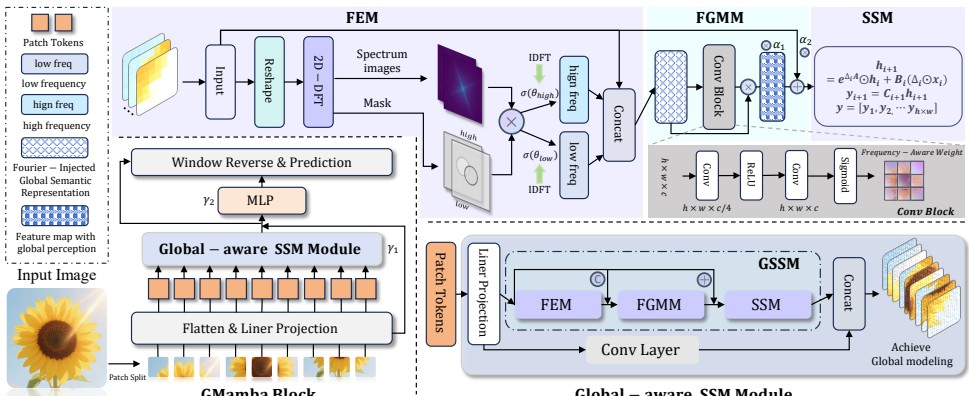

Figure 2: Overview of the GMamba Block. The input image is first segmented into multiple patches and projected into a token sequence, which is then processed by the GSSM module. Unlike vanilla SSM, GSSM exhibits true global perception capabilities through DFT-based pre-modulation using FEM and FGMM, while providing linear logarithmic complexity. Finally, MLP is applied for further semantic processing.

## 2.3 GMAMBA OVERALL ARCHITECTURE

Based on the theoretical exploration in Sec. 2.2 on endowing SSMs with global perception through frequency-guided modulation, we implemented the design of GMamba and GSSM, whose overall architecture is shown in Fig. 2.

Specifically, for an input image $X \in \mathbb{R}^{H \times W \times C}$, we first partition it into patches of size $P \times P$, obtaining $N = \frac{H \times W}{P^2}$ patch tokens, represented as $\{x_1, x_2, \ldots, x_N\}$, where each $x_i \in \mathbb{R}^{P^2 \times C}$. Subsequently, through linear projection, these tokens are mapped to the feature space, yielding the projected feature sequence $Z \in \mathbb{R}^{N \times D}$. The feature sequence is then fed into the GSSM module for global contextual modeling, with its output features further processed through a Multi-layer Perceptron (MLP) to extract semantic information, and finally restored to spatial dimensions $\mathbb{R}^{H \times W \times D}$ through window reconstruction operations. The overall process of GMamba Block is: $Z = \text{WindowPartition}(X)$, $Z = Z + \text{DropPath}(\gamma_1 \cdot \text{GSSM}(\text{Norm}_1(Z)))$, $Z = Z + \text{DropPath}(\gamma_2 \cdot \text{MLP}(\text{Norm}_2(Z)))$, and $\hat{X} = \text{WindowReverse}(Z)$, where $\gamma_1$ and $\gamma_2$ are learnable scaling factors for GSSM and MLP branches, respectively, $Norm_1$ and $Norm_2$ represent layer normalization operations, and $DropPath$ is stochastic depth regularization.

## 2.4 GSSM MODULE

GSSM consists of two key modules: Frequency Encoding Module (FEM) and Frequency-Guided Modulation Module (FGMM), as illustrated in Fig. 2. FEM applies 2D-DFT to input features, extracting high- and low-frequency components, then adaptively recalibrates using learnable weights. This injects global semantics while guiding the model to focus on critical regions. FGMM leverages modulation coefficients $\alpha_1$ and $\alpha_2$ to adaptively interact frequency-domain and spatial features before spatial modeling, enabling frequency-guided modulation. Resulting features are subsequently processed by SSM. The appendix provides detailed algorithmic flow of GSSM, along with analysis of its linear-logarithmic computational complexity.

## 3 EXPERIMENTS

In this section, we conduct experiments on five datasets across multiple backbones and tasks (semantic segmentation, object detection, instance segmentation) to validate GMamba's effectiveness. We compare it with existing global modeling methods and analyze the design from various perspectives.

### 3.1 SEMANTIC SEGMENTATION ON VAIHINGEN, POTSDAM, LOVEDA, AND UAVID

Our semantic segmentation experiments are conducted on four remote sensing datasets: Vaihingen (Rottensteiner et al., 2020), Potsdam (Rottensteiner et al., 2020), LoveDA (Wang et al., 2021), and UAVid (Lyu et al., 2020). Detailed descriptions of each dataset are provided in the appendix.

**Backbones.** We use the ResNet (He et al., 2016), Swin Transformer (Liu et al., 2021), and ConvNeXt (Woo et al., 2023) families for experiments, covering lightweight CNNs, larger CNNs, and global modeling architectures. Specifically, we choose ResNet34 (Koonce, 2021), Swin Transformer-Tiny (Swin(T)), and ConvNeXt-Small (ConvNeXt(S)) as the backbones.

Table 1: Performance comparison on Vaihingen, Potsdam, LoveDA and UAVid datasets. GMamba consistently improves performance across different backbone architectures.

| Category | Vaihingen Method | Vaihingen | | | Potsdam Method | Potsdam | | |
|---|---|---|---|---|---|---|---|---|
| | | mIoU(%) | mFl(%) | OA(%) | | mIoU(%) | mFl(%) | OA(%) |
| Existing Methods | UNetFormer (Wang et al., 2022) | 83.14 | 90.57 | 90.99 | UNetFormer (Wang et al., 2022) | 86.14 | 92.42 | 91.21 |
| | CMTFNet (Wu et al., 2023) | 83.74 | 90.95 | 91.15 | CMTFNet (Wu et al., 2023) | 86.05 | 92.36 | 90.92 |
| | CGGLNet (Ni et al., 2024) | 83.89 | 91.12 | 91.47 | CGGLNet (Ni et al., 2024) | 87.11 | 92.96 | 91.58 |
| | SFFNet (Yang et al., 2024) | 84.38 | 91.32 | 91.36 | SFFNet (Yang et al., 2024) | 86.93 | 92.66 | 91.72 |
| | MCSNet (Chen et al., 2024b) | 84.26 | 91.36 | 91.97 | MCSNet (Chen et al., 2024b) | 86.92 | 92.89 | 91.40 |
| | AFENet (Gao et al., 2025b) | 84.55 | 91.54 | 91.67 | AFENet (Gao et al., 2025b) | 87.50 | 93.24 | 92.03 |
| Baseline Networks | UNet-ResNet34 | 81.65 | 89.24 | 91.86 | UNet-ResNet34 | 83.90 | 90.70 | 89.12 |
| | UNet-Swin(T) | 82.44 | 89.75 | 92.18 | UNet-Swin(T) | 84.69 | 91.16 | 89.73 |
| | UNet-ConvNeXt(S) | 83.11 | 90.19 | 92.30 | UNet-ConvNeXt(S) | 85.09 | 91.40 | 89.86 |
| Ours (+GMamba) | UNet-ResNet34-GMamba | **84.74**(↑3.09) | **91.56**(↑2.32) | **93.72**(↑1.86) | UNet-ResNet34-GMamba | **86.21**(↑2.31) | **92.48**(↑1.78) | **91.06**(↑1.94) |
| | UNet-Swin(T)-GMamba | **84.83**(↑2.39) | **91.61**(↑1.86) | **93.65**(↑1.47) | UNet-Swin(T)-GMamba | **86.94**(↑2.25) | **92.90**(↑1.74) | **91.40**(↑1.67) |
| | UNet-ConvNeXt(S)-GMamba | **86.00**(↑2.89) | **92.31**(↑2.12) | **93.99**(↑1.69) | UNet-ConvNeXt(S)-GMamba | **87.71**(↑2.62) | **93.34**(↑1.94) | **91.83**(↑1.97) |

| Category | LoveDA Method | LoveDA | | | UAVid Method | mIoU(%) |
|---|---|---|---|---|---|---|
| | | mIoU(%) | mFl(%) | OA(%) | | |
| Existing Methods | DeepLabv3+ (Yurtkulu et al., 2019) | 49.11 | 64.97 | 66.65 | UNetFormer (Wang et al., 2022) | 67.80 |
| | ST-UNet (He et al., 2022) | 54.43 | - | 67.18 | MFIN (Chen & Luo, 2024) | 69.70 |
| | MaskFormer (Cheng et al., 2022) | 50.79 | 66.87 | 70.82 | UrbanSSF (Ni et al., 2024) | 69.60 |
| | Mask2Former (Cheng et al., 2022) | 53.14 | 68.82 | 71.93 | MA-DBFAN (Yue et al., 2024) | 70.01 |
| | MMT (Xu et al., 2023) | 54.92 | 69.64 | 72.34 | CAGNet (Wang et al., 2024) | 69.00 |
| | GLOTS (Liu et al., 2023) | 55.63 | - | 67.42 | UMFormer (Li et al., 2025b) | 67.60 |
| Baseline Networks | UNet-ResNet34 | 50.37 | 66.09 | 66.88 | UNet-ResNet34 | 64.25 |
| | UNet-Swin(T) | 51.05 | 66.79 | 67.82 | UNet-Swin(T) | 65.73 |
| | UNet-ConvNeXt(S) | 54.97 | 69.98 | 71.80 | UNet-ConvNeXt(S) | 69.15 |
| Ours (+GMamba) | UNet-ResNet34-GMamba | **52.80**(↑2.43) | **68.31**(↑2.22) | **69.92**(↑3.04) | UNet-ResNet34-GMamba | **66.38**(↑2.13) |
| | UNet-Swin(T)-GMamba | **54.00**(↑2.95) | **69.33**(↑2.54) | **70.46**(↑2.64) | UNet-Swin(T)-GMamba | **67.62**(↑1.89) |
| | UNet-ConvNeXt(S)-GMamba | **57.17**(↑2.20) | **72.17**(↑2.19) | **74.37**(↑2.57) | UNet-ConvNeXt(S)-GMamba | **70.22**(↑1.07) |

**Experimental Setup.** In experiments, we select UNet as the baseline network and insert GMamba in a residual connection manner at each stage of the network encoder and decoder, with a total of 7 GMamba modules inserted. We compare against state-of-the-art global modeling methods: (1) attention-based: Swin Transformer (Swin) (Liu et al., 2021), Swin Transformer V2 (SwinV2) (Liu et al., 2022), and (2) SSM-based: Vim (Zhu et al., 2024a), VMamba (Liu et al., 2024b), TinyVim (Ma et al., 2024). All methods use identical insertion strategies. For fair comparison, we employ publicly available code and common training/testing settings from GeoSeg (Wang et al., 2022). All models use identical configurations: batch size, training epochs, etc. Details are in the appendix.

Table 2: Performance comparison of GMamba and other global modeling modules on the Vaihingen dataset across different backbones.

| Model Variant | Params (M) | FLOPs (G) | mIoU (%) | mF1 (%) | OA (%) |
|---|---|---|---|---|---|
| UNet-ResNet34 (Baseline) | 25.33 | 30.32 | 81.65 | 89.24 | 91.86 |
| + Swin (×7) | 35.81 | 43.45 | 83.24 (↑1.59) | 90.63 (↑1.39) | 93.08 (↑1.22) |
| + SwinV2 (×7) | 35.86 | 43.46 | 83.10 (↑1.45) | 90.54 (↑1.30) | 93.04 (↑1.18) |
| + ViM (×7) | 28.20 | 35.81 | 83.02 (↑1.37) | 90.51 (↑1.27) | 92.93 (↑1.07) |
| + VMamba (×7) | 32.45 | 38.49 | 83.24 (↑1.59) | 90.62 (↑1.38) | 93.04 (↑1.18) |
| + TinyViM (×7) | 31.19 | 36.51 | 83.17 (↑1.52) | 90.59 (↑1.35) | 92.99 (↑1.13) |
| + Mamba Version (×7) | 34.87 | 39.78 | 83.20 (↑1.55) | 90.62 (↑1.38) | 93.00 (↑1.14) |
| + Spatial Mamba (×7) | 30.92 | 35.72 | 82.90 (↑1.25) | 90.40 (↑1.16) | 92.85 (↑0.99) |
| + FreqMamba (×7) | 31.98 | 36.21 | 83.00 (↑1.35) | 90.50 (↑1.26) | 92.90 (↑1.04) |
| + Group Mamba (×7) | 29.72 | 35.15 | 82.99 (↑1.34) | 90.41 (↑1.17) | 92.86 (↑1.00) |
| + GMamba (×7) (Ours) | 30.96 | 36.30 | **84.74** (↑3.09) | **91.56** (↑2.32) | **93.72** (↑1.86) |
| UNet-Swin(T) (Baseline) | 36.48 | 44.46 | 82.44 | 89.75 | 92.18 |
| + Swin (×7) | 60.01 | 73.27 | 83.70 (↑1.26) | 90.92 (↑1.17) | 93.12 (↑0.94) |
| + SwinV2 (×7) | 60.07 | 73.28 | 83.73 (↑1.29) | 90.94 (↑1.19) | 93.19 (↑1.01) |
| + ViM (×7) | 42.83 | 57.99 | 83.62 (↑1.18) | 90.88 (↑1.13) | 93.05 (↑0.87) |
| + VMamba (×7) | 52.06 | 61.99 | 84.07 (↑1.63) | 91.14 (↑1.39) | 93.32 (↑1.14) |
| + TinyViM (×7) | 49.38 | 58.01 | 84.04 (↑1.60) | 91.13 (↑1.38) | 93.28 (↑1.10) |
| + Mamba Version (×7) | 52.47 | 62.95 | 83.68 (↑1.24) | 90.90 (↑1.15) | 93.10 (↑0.92) |
| + Spatial Mamba (×7) | 48.93 | 56.88 | 83.40 (↑0.96) | 90.70 (↑0.95) | 92.92 (↑0.74) |
| + FreqMamba (×7) | 49.95 | 57.98 | 83.55 (↑1.11) | 90.82 (↑1.07) | 93.00 (↑0.82) |
| + Group Mamba (×7) | 46.31 | 55.15 | 83.51 (↑1.07) | 90.80 (↑1.05) | 92.97 (↑0.79) |
| + GMamba (×7) (Ours) | 49.13 | 57.81 | **84.83** (↑2.39) | **91.61** (↑1.86) | **93.65** (↑1.47) |
| UNet-ConvNeXt(S) (Baseline) | 58.42 | 68.88 | 83.11 | 90.19 | 92.30 |
| + Swin (×7) | 81.95 | 97.69 | 84.82 (↑1.71) | 91.59 (↑1.40) | 93.57 (↑1.27) |
| + SwinV2 (×6) | 81.77 | 93.61 | 84.36 (↑1.25) | 91.31 (↑1.12) | 93.32 (↑1.02) |
| + ViM (×7) | 64.76 | 78.09 | 84.24 (↑1.13) | 91.22 (↑1.03) | 93.37 (↑1.07) |
| + VMamba (×7) | 73.99 | 86.41 | 84.56 (↑1.45) | 91.45 (↑1.26) | 93.56 (↑1.26) |
| + TinyViM (×7) | 71.31 | 82.43 | 84.38 (↑1.27) | 91.33 (↑1.14) | 93.41 (↑1.11) |
| + Mamba Version (×7) | 74.87 | 88.47 | 84.80 (↑1.69) | 91.55 (↑1.36) | 93.55 (↑1.25) |
| + Spatial Mamba (×7) | 70.93 | 84.92 | 84.50 (↑1.39) | 91.32 (↑1.13) | 93.35 (↑1.05) |
| + FreqMamba (×7) | 71.95 | 85.98 | 84.60 (↑1.49) | 91.42 (↑1.23) | 93.42 (↑1.12) |
| + Group Mamba (×7) | 68.20 | 79.60 | 84.56 (↑1.45) | 91.44 (↑1.25) | 93.38 (↑1.08) |
| + GMamba (×7) (Ours) | 71.06 | 85.66 | **86.00** (↑2.89) | **92.31** (↑2.12) | **93.99** (↑1.69) |

**Results Comparison.** Table 1 presents semantic segmentation performance results of GMamba on four remote sensing datasets using different backbone networks. GMamba achieves consistent mIoU improvements across all datasets and backbones, demonstrating universality and plug-and-play characteristics for both CNN and Transformer-based backbones. GMamba particularly excels when paired with the powerful ConvNeXt(S) backbone, surpassing existing methods on all datasets. To

validate GMamba's effectiveness, we compare it with various global modeling modules using UNet baseline on Vaihingen dataset. Table 2 shows that while Transformer and Mamba-based modules improve accuracy on lightweight ResNet34, Transformer variants (Swin, SwinV2) introduce substantial parameters and FLOPs due to quadratic complexity. Mamba variants (Vim, VMamba, TinyVim) show limited global modeling capability with modest gains. In contrast, GMamba achieves superior accuracy-efficiency trade-off. On global modeling backbones (Swin(T)), GMamba extracts fine-grained global information to enhance performance. On complex CNN backbones (ConvNeXt(S)), it achieves optimal results. Complexity and qualitative analysis is provided in Appendix.

## 3.2 OBJECT DETECTION AND INSTANCE SEGMENTATION ON MS-COCO

To verify the effectiveness of the GMamba Block on other downstream tasks, we conduct experiments on object detection and instance segmentation using the 2017 MS-COCO dataset, which contains 118,000 training and 5,000 validation images, covering 80 object categories.

**Experimental Setup.** We use the MMDetection (Chen et al., 2019) framework and conduct experiments on the MS-COCO dataset. For object detection, we adopt pre-trained ResNet50 and Swin(T) as the backbone networks of the detectors and employ Faster R-CNN (Ren et al., 2016) and Mask R-CNN (He et al., 2017) as the detection models, with Feature Pyramid Network (FPN) (Lin et al., 2017) as the neck structure. For instance segmentation, we use Mask R-CNN with Swin(T) as the backbone and FPN as the neck. We insert the GMamba module into the first three stages of the backbone through residual connections to provide global information. All comparison modules (Swin, SwinV2, Vim, VMamba, and TinyVim) adopt the same insertion strategy and training configuration. Further experimental details and insertion strategies are provided in the appendix.

Table 3: Comparison of GMamba and other global modeling modules inserted into the first three stages of different backbones (ResNet50 / Swin(T)) on the MS-COCO object detection task with Faster R-CNN and Mask R-CNN.

| Model Variant | AP (%) | $AP_{50}$ (%) | $AP_{75}$ (%) | $AP_S$ (%) | $AP_M$ (%) | $AP_L$ (%) | Params (M) | FLOPs (G) |
|---|---|---|---|---|---|---|---|---|
| | | | | *Faster R-CNN* | | | | |
| Baseline (ResNet50) | 37.2 | 57.8 | 40.4 | 21.5 | 40.6 | 48.0 | 43.80(23.5) | 207.07(76.50) |
| + Swin (×3) | 38.0 | 59.1 | 41.0 | **23.0** | 41.5 | 48.4 | 76.70(56.4) | 210.57(80.00) |
| + SwinV2 (×3) | 37.7 | 58.7 | 40.9 | 21.5 | 41.3 | 48.8 | 76.90(56.6) | 210.37(79.80) |
| + TinyViM (×3) | 37.6 | 58.6 | 40.6 | 21.6 | 41.3 | 48.4 | 61.40(41.1) | 208.87(78.30) |
| + VMamba (×3) | 37.6 | 58.8 | 40.8 | 21.5 | 41.4 | 48.8 | 65.00(44.7) | 209.67(79.10) |
| + Mamba Version (×3) | 37.7 | 58.9 | 41.0 | 21.6 | 41.4 | 48.5 | 65.40(45.1) | 209.97(79.40) |
| + Spatial Mamba (×3) | 37.5 | 58.6 | 40.8 | 21.4 | 41.2 | 48.3 | 64.30(44.0) | 209.27(78.70) |
| + FreqMamba (×3) | 37.6 | 58.7 | 40.9 | 21.5 | 41.3 | 48.4 | 64.90(44.6) | 209.57(79.00) |
| + Group Mamba (×3) | 37.5 | 58.5 | 40.5 | 21.3 | 41.0 | 48.5 | 64.80(44.5) | 209.37(78.80) |
| + GMamba (×3) (Ours) | **38.5** | **59.6** | **42.2** | 22.1 | **42.0** | **49.9** | 61.40(41.1) | 210.22(79.65) |
| Baseline (Swin (T)) | 41.6 | 64.0 | 45.2 | 25.7 | 44.8 | 55.4 | 45.15(27.5) | 213.00(86.10) |
| + GMamba (×3) (Ours) | **42.9** | **65.6** | **46.9** | **27.1** | **46.4** | **56.4** | 55.25(37.6) | 216.15(89.25) |
| | | | | *Mask R-CNN* | | | | |
| Baseline (ResNet50) | 37.5 | 58.1 | 41.3 | 21.2 | 40.9 | 48.3 | 46.45(23.5) | 260.14(76.50) |
| + Swin (×3) | 38.4 | 59.2 | 41.7 | 21.6 | 42.2 | 49.3 | 79.35(56.4) | 263.64(80.00) |
| + SwinV2 (×3) | 38.3 | 59.1 | 41.7 | 22.2 | 41.9 | 49.6 | 79.55(56.6) | 263.44(79.80) |
| + TinyViM (×3) | 38.2 | 58.8 | 41.6 | 21.5 | 41.6 | 49.9 | 64.05(41.1) | 261.94(78.30) |
| + VMamba (×3) | 38.4 | 59.1 | 41.9 | 22.3 | 41.8 | 49.8 | 67.65(44.7) | 262.74(79.10) |
| + Mamba Version (×3) | 38.2 | 59.8 | 42.2 | 22.3 | 42.0 | 49.2 | 68.05(45.1) | 263.04(79.40) |
| + Spatial Mamba (×3) | 38.0 | 59.4 | 42.0 | 22.0 | 41.8 | 49.0 | 66.95(44.0) | 262.34(78.70) |
| + FreqMamba (×3) | 38.1 | 59.6 | 42.1 | 22.1 | 41.9 | 49.1 | 67.55(44.6) | 262.64(79.00) |
| + Group Mamba (×3) | 38.3 | 59.0 | 41.8 | 22.6 | 41.4 | 50.2 | 67.45(44.5) | 262.44(78.80) |
| + GMamba (×3) (Ours) | **39.1** | **60.1** | **42.8** | **23.3** | **42.5** | **50.3** | 64.05(41.1) | 263.29(79.65) |
| Baseline (Swin(T)) | 42.2 | 64.7 | 46.0 | 26.7 | 45.5 | 55.6 | 47.79(27.5) | 266.00(86.10) |
| + GMamba (×3) (Ours) | **43.7** | **66.4** | **48.0** | **27.5** | **47.3** | **57.0** | 57.89(37.6) | 269.15(89.25) |

Table 4: Comparison of GMamba and other global modeling modules with Mask R-CNN (Swin(T) backbone) on MS-COCO instance segmentation

| Model Variant | AP (%) | $AP_{50}$ (%) | $AP_{75}$ (%) | $AP_S$ (%) | $AP_M$ (%) | $AP_L$ (%) | Params (M) | FLOPs (G) |
|---|---|---|---|---|---|---|---|---|
| | | | | *Mask R-CNN* | | | | |
| Baseline (Swin(T)) | 38.7 | 61.3 | 41.5 | 20.2 | 41.6 | 56.7 | 47.79(27.5) | 266.00(86.10) |
| + Swin (×3) | 38.4 | 59.2 | 41.7 | **21.6** | 42.2 | 49.3 | 66.49(46.2) | 269.50(89.60) |
| + SwinV2 (×3) | 39.1 | 61.9 | 42.0 | 21.3 | 42.3 | 57.4 | 67.49(46.3) | 269.30(89.40) |
| + TinyViM (×3) | 38.2 | 58.8 | 41.6 | 21.5 | 41.6 | 49.9 | 58.09(37.8) | 267.80(87.90) |
| + VMamba (×3) | 38.9 | 61.8 | 41.9 | 19.8 | 41.7 | 56.9 | 60.59(40.3) | 268.60(88.70) |
| + Mamba Version (×3) | 39.0 | 61.2 | 42.0 | 21.0 | 41.6 | 57.0 | 66.39(46.1) | 269.50(89.60) |
| + Spatial Mamba (×3) | 38.6 | 60.7 | 41.6 | 20.6 | 41.3 | 56.5 | 66.19(45.9) | 269.20(89.30) |
| + FreqMamba (×3) | 39.2 | 61.4 | 42.2 | 21.1 | 41.7 | 57.2 | 66.29(46.0) | 269.30(89.40) |
| + Group Mamba (×3) | 39.1 | 61.9 | 41.9 | 20.2 | 41.9 | 57.2 | 63.79(43.5) | 268.90(89.00) |
| + GMamba (×3) (Ours) | **39.8** | **62.7** | **42.8** | 20.1 | **42.7** | **58.0** | 57.89(37.6) | 269.15(89.25) |

**Results Comparison.** From the results in Tables 3 and 4, GMamba provides consistent performance improvements on both object detection and instance segmentation tasks. On Faster R-CNN and Mask R-CNN with a pre-trained ResNet-50 backbone, GMamba achieves an $AP_L$ improvement of 1.9% and 2.0% over baseline, demonstrating its strong global modeling capability. Our method also significantly outperforms other global modeling modules when integrated with Swin(T). Due to space constraints, the comparison results between GMamba and other global modeling modules

using the Swin(T) backbone on object detection are provided in the appendix. GMamba exhibits strong performance on instance segmentation, achieving the best results on most AP metrics.

## 3.3 ABLATION STUDIES

Finally, we conduct extensive ablation studies on the Vaihingen dataset using UNet with ConvNeXt-Small as the baseline to validate the effectiveness of the GSSM module design.

Table 5: Ablation Study of GSSM Components

| Model Variant | GSSM Module | | | Performance | | | | |
|---|---|---|---|---|---|---|---|---|
| | FEM | FGMM | SSM | Params (M) | FLOPs (G) | mIoU (%) | mF1 (%) | OA (%) |
| Baseline (UNet-ConvNeXt-Small) | – | – | – | 58.42 | 68.88 | 83.11 | 90.19 | 92.30 |
| + SSM only | – | – | ✓ | 68.31 | 82.00 | 84.01 | 91.01 | 93.38 |
| + FEM + SSM | ✓ | – | ✓ | 70.80 | 85.50 | 85.30 | 91.89 | 93.81 |
| (DFT)+ FGMM + SSM | – | ✓ | ✓ | 68.62 | 82.24 | 84.79 | 91.58 | 93.51 |
| + GSSM Module (All) | ✓ | ✓ | ✓ | 71.06 | 85.66 | **86.00** | **92.31** | **93.99** |

Table 6: Ablation Study on Scan Strategies in GSSM

| Scan Strategy | Params (M) | FLOPs (G) | mIoU (%) | mF1 (%) | OA (%) |
|---|---|---|---|---|---|
| Bidirectional Scan | 75.75 | 88.50 | 85.73 | 92.15 | 93.88 |
| Four-way Scan | 78.50 | 91.00 | 85.98 | 92.30 | 93.98 |
| Single-direction (Ours) | 71.06 | 85.66 | **86.00** | **92.31** | **93.99** |

Table 7: Effect of Replacing Global Modeling Module in GMamba

| GMamba Variant | Params (M) | FLOPs (G) | mIoU (%) | mF1 (%) | OA (%) |
|---|---|---|---|---|---|
| GMamba w/ Self-Attention | 74.87 | 91.20 | 84.78 | 91.57 | 93.62 |
| GMamba w/ VSSM (Mamba) | 68.31 | 82.00 | 84.01 | 91.01 | 93.38 |
| GMamba w/ GSSM (Ours) | 71.06 | 85.66 | 86.00 | 92.31 | 93.99 |

**Effectiveness of GSSM components.** As shown in Table 5, SSM alone achieves +0.90% mIoU improvement, demonstrating effective global information capture. Adding FEM significantly boosts performance to 85.30% mIoU, validating its global semantic injection capability. Replacing FEM with simple DFT and adding FGMM drops performance to 84.79% mIoU, indicating the necessity of adaptive mechanisms. The complete GSSM module achieves optimal performance with reasonable computational overhead, validating the theoretical design.

**The Impact of Scan Strategies in GSSM.** We modified GSSM to incorporate bidirectional and four-way scanning, as shown in Table 6. The results demonstrate that GSSM can achieve genuine global perception through frequency-domain modulation using only unidirectional scanning, without the need for more complex scanning mechanisms. Introducing additional scanning strategies merely increases computational overhead.

**Effectiveness of GSSM for global modeling.** As shown in Table 7, our GSSM not only demonstrates superiority in theoretical analysis but is also empirically shown to outperform existing global modeling modules. Although it introduces slightly higher computational complexity compared to the linear-complexity VSSM, it achieves a substantial performance improvement. Compared with standard self-attention, GSSM maintains significantly lower complexity while delivering the best overall performance.

**More Ablations.** In the Appendix, we provide additional ablation studies, including: (1) the impact of applying GMamba at different layer positions; (2) the analysis of GMamba's receptive field; (3) the effect of frequency-domain information; (4) the robustness evaluation; (5) the efficiency analysis of GSSM; and (6) the impact of different frequency-domain integration strategies, among others.

## 4 CONCLUSION AND DISCUSSION

This paper first formulates a rigorous mathematical definition of image global modeling, laying the foundation for designing globally-aware models with stronger theoretical grounding and interpretability. We establish a comprehensive theoretical framework for frequency-domain modulation, resulting in an SSM architecture that is explicit, uniform, non-causal, and mathematically well-defined, making it more suitable for the global modeling of images. Guided by this theory, we design the GSSM module and, based on it, construct the plug-and-play GMamba module. By introducing a DFT-based global attribute modulation mechanism, the SSM overcomes the lack of an explicit non-causal global perception mechanism. Both theoretical analysis and empirical evaluation demonstrate that this module exhibits excellent global modeling capability and strong adaptability across multiple tasks. See the appendix for details on LLM usage.

ACKNOWLEDGEMENTS

This work was supported in part by the National Natural Science Foundation of China (Grant Nos. 62495091, 62125305, and 52508037).

ETHICS STATEMENT

This study does not involve human subjects, personally identifiable information, or sensitive user data. The experiments are conducted on publicly available datasets that have been widely used in prior research. We adhere to the dataset licenses and usage agreements and do not raise any additional ethical concerns. The proposed methods are intended solely for academic research and do not introduce foreseeable harmful applications.

REPRODUCIBILITY STATEMENT

We have taken several measures to ensure the reproducibility of our experiments. All model architectures, training procedures, and hyperparameter settings are described in comprehensive detail. Dataset preprocessing strictly follows the official guidelines and standard protocols. All theoretical derivations and proofs are provided in the appendix. To further support reproducibility, we provide anonymized source code in the supplementary materials.

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

APPENDIX

In Sec. A, we describe the use of LLMs. In Sec. B, we discuss the related work. In Sec. C, we present the complete derivation of the discrete-time SSM frequency-domain transfer function. In Sec. D, we provide a full proof of the global properties of the DFT, and in Sec. E, we give a detailed proof that the 2D-DFT satisfies our proposed definition for global image modeling. In Sec. F and Sec. G, we detail the algorithm of the GSSM module and analyze the computational complexity of GMamba. In Sec. H, we describe the datasets and experimental settings in detail. In Sec. I, we provide additional experiments on semantic segmentation tasks, including (1) comparisons of model complexity, (2) ablation studies for integrating GMamba at different stages, and (3) visual result analysis. In Sec. J, we present further experiments on object detection and instance segmentation, including (1) comprehensive comparisons for object detection, (2) ablation studies for inserting GMamba at different stages, and (3) visualization analysis. In Sec. K, we conduct additional analyses on the GSSM module. Finally, in Sec. L, we analyzed the failure cases.

## A    THE USE OF LARGE LANGUAGE MODELS

To improve the quality of the manuscript, LLMs were used to assist with writing and editing, limited solely to language polishing. The authors remain fully responsible for the content.

## B    RELATED WORK

### B.1    STATE SPACE MODELS IN VISUAL TASKS

State Space Models offer linear computational complexity and efficiently capture long-range dependencies, making them a promising alternative to Transformers in computer vision. However, SSMs rely on autoregressive formulations suited for causal sequences, a property that image data inherently lacks (Yu & Wang, 2025). To address this, several improvements (Liu et al., 2024a; Chen et al., 2024a; Cheng et al., 2024; Shi et al., 2025b; Deng & Gu, 2024; Xu et al., 2025; Hui et al., 2025; Shaker et al., 2025a; Ma et al., 2026) have been proposed.

To better capture 2D spatial context, many works have focused on modifying the scanning mechanism. For instance, Vim (Zhu et al., 2024a) uses bidirectional SSMs to enhance spatial understanding by modeling forward and backward directions, but at a high computational cost and with limited fusion effectiveness. Mamba Vision (Hatamizadeh & Kautz, 2025) preserves the one-directional SSM formulation but incorporates self-attention to model global image dependencies, which also increases the computational complexity. VMamba (Liu et al., 2024b) introduces a two-dimensional selective scanning mechanism to capture context from multiple directions. Further addressing the 2D spatial structure challenge, Spatial-Mamba (Xiao et al., 2025) introduces a structure-aware state fusion equation, leveraging dilated convolutions to directly enhance spatial context modeling within the state space. Similarly, GroupMamba (Shaker et al., 2025b) employs grouped scanning mechanisms to improve the representation of spatial relationships.

Despite these advances, existing methods still face limitations. Approaches like VMamba, Spatial-Mamba, and GroupMamba attempt to capture 2D context by complicating the spatial scanning or fusion mechanisms, which can increase computational overhead (Zhu et al., 2024b) and still rely on autoregressive mechanisms to progressively build global representations. In contrast, our GSSM offers a simpler, more efficient design by adopting unidirectional modeling and integrating frequency-domain global modeling to guide state selection and updates, enabling true global awareness tailored for image tasks.

### B.2    FREQUENCY LEARNING

The Discrete Fourier Transform (Bone et al., 1986; Pei & Yeh, 1998; Winograd, 1978; Sundararajan, 2001; Feng et al., 2023) is a classical technique for analyzing the frequency-domain characteristics of digital signals, providing a global representation. Recent studies (Mao et al., 2023; Liu et al., 2025; Cai et al., 2021; Gao et al., 2024; Feng et al., 2024) have explored combining frequency and spatial information for visual tasks. For example, AFENet (Gao et al., 2025a) introduces an adaptive

frequency–spatial interaction module, and SFFNet (Yang et al., 2024) fuses frequency components with CNNs. Other works propose novel frequency-based operators. SPANet (Yun et al., 2023) designs a frequency-balancing token mixer using spectral modulation. FAD (Shi et al., 2025a), addressing cross-domain learning, proposes a Frequency Diversion Adapter that partitions features into frequency bands and adapts each band separately. Similarly, FreqMamba (Zou et al., 2024), designed for image deraining, introduces a "triple interaction structure" with parallel spatial Mamba, "frequency band Mamba," and Fourier global modeling modules.

Building on this foundation, we further investigate how to integrate frequency-domain information into SSMs. Most existing approaches—such as dual-branch frequency-domain structures (Luan et al., 2025; Tan et al., 2024; Bo et al., 2025), complex parallel architectures like FreqMamba (Zou et al., 2024), newly designed frequency mixers (Yun et al., 2023), or multi-band adapters (Shi et al., 2025a)—enhance global perception through post-hoc fusion or external block-level modulation. While such external adaptations are effective, the frequency-domain information merely serves as a supplementary signal to spatial-domain features and cannot influence the underlying modeling mechanism. In contrast, our GSSM is grounded in the proposed theoretical framework of frequency-modulated SSMs. We introduce frequency-domain information before the modeling process and allow it to directly participate in the state-update dynamics of the SSM itself. This enables the model to acquire a global perspective during the core modeling stage, rather than simply enriching the final feature representation.

## C  DERIVATION OF DISCRETE-TIME SSM FREQUENCY-DOMAIN TRANSFER FUNCTION

We provide a detailed derivation of the frequency-domain transfer function for discrete-time SSMs.

Consider a discrete-time state-space model defined by:

$$x_t = \bar{A}x_{t-1} + \bar{B}u_t \tag{19}$$
$$y_t = Cx_t \tag{20}$$

where $\bar{A} = e^{A\Delta}$ is the discrete-time state transition matrix, $\bar{B} = A^{-1}(e^{A\Delta} - I)B$ is the discrete-time input matrix, $C$ is the output matrix, and the system is causal and stable with $\rho(\bar{A}) < 1$.

Our objective is to derive the frequency-domain transfer function:

$$H(\omega) = C(e^{j\omega}I - \bar{A})^{-1}\bar{B} \tag{21}$$

### C.1  DERIVATION

Taking the Z-transform of Eq. 19 with zero initial conditions ($x_0 = 0$):

$$zX(z) = \bar{A}X(z) + \bar{B}U(z) \tag{22}$$

Rearranging terms:
$$(zI - \bar{A})X(z) = \bar{B}U(z) \tag{23}$$

Solving for $X(z)$:
$$X(z) = (zI - \bar{A})^{-1}\bar{B}U(z) \tag{24}$$

Taking the Z-transform of Eq. 20:
$$Y(z) = CX(z) \tag{25}$$

Substituting Eq. 24:
$$Y(z) = C(zI - \bar{A})^{-1}\bar{B}U(z) \tag{26}$$

The Z-domain transfer function is defined as:
$$H(z) = \frac{Y(z)}{U(z)} = C(zI - \bar{A})^{-1}\bar{B} \tag{27}$$

By applying $z = e^{j\omega}$ to Eq. 27, we obtain the frequency-domain transfer function:

$$H(\omega) = H(z)\Big|_{z=e^{j\omega}} = C(e^{j\omega}I - \bar{A})^{-1}\bar{B}. \tag{28}$$

### C.2 STABILITY ANALYSIS

The stability condition $\rho(\bar{A}) < 1$ ensures that the transfer function $H(\omega)$ is well-defined for all $\omega \in [-\pi, \pi]$.

If $\rho(\bar{A}) < 1$, then the matrix $(I - \bar{A}e^{-j\omega})$ is invertible for all $\omega \in \mathbb{R}$. Let $\lambda$ be any eigenvalue of $\bar{A}$ with $|\lambda| < 1$. For any $\omega \in \mathbb{R}$, we have $|e^{-j\omega}| = 1$, so $|\lambda e^{-j\omega}| = |\lambda| < 1$. This implies that $\lambda e^{-j\omega} \neq 1$ for any eigenvalue $\lambda$ of $\bar{A}$. Therefore, the matrix $(I - \bar{A}e^{-j\omega})$ has no zero eigenvalues and is invertible.

### C.3 FINAL RESULT

The frequency-domain transfer function for a causal and stable discrete-time SSM is:

$$H(\omega) = C(e^{j\omega}I - \bar{A})^{-1}\bar{B}. \tag{29}$$

## D COMPLETE PROOF OF THE GLOBAL PROPERTIES OF THE DFT

To intuitively demonstrate the global properties of the DFT, we visualize the global perception of DFT as illustrated in Fig. 3: when a minor local perturbation is applied to a time-domain signal, its frequency spectrum undergoes extensive global changes; conversely, local modifications in the frequency domain also induce holistic fluctuations throughout the time-domain signal.

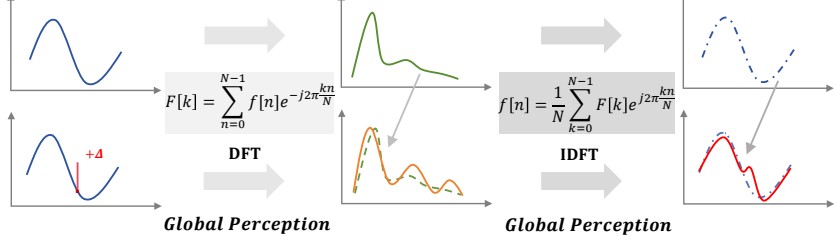

Figure 3: A small perturbation in the time-domain signal leads to global changes in the frequency spectrum, and vice versa, illustrating the global sensitivity of both the DFT and its inverse.

Moreover, we provide a rigorous mathematical proof demonstrating that the DFT possesses a global property: a local perturbation in one domain (time or frequency) induces non-zero responses at all indices in the dual domain.

Let $f[n] \in \mathbb{C}^N$ be a discrete signal of length $N$. The DFT and its inverse are defined as follows:

$$F[k] = \sum_{n=0}^{N-1} f[n] \cdot e^{-i\frac{2\pi}{N}kn}, \quad k = 0, 1, \ldots, N-1 \tag{30}$$

$$f[n] = \frac{1}{N}\sum_{k=0}^{N-1} F[k] \cdot e^{i\frac{2\pi}{N}kn}, \quad n = 0, 1, \ldots, N-1 \tag{31}$$

### D.1 LOCAL PERTURBATION IN TIME AFFECTS ALL FREQUENCIES

We introduce a small additive perturbation $\Delta f \in \mathbb{C}$ at time index $n_0$. The perturbed signal becomes:

$$f'[n] = \begin{cases} f[n] + \Delta f, & n = n_0 \\ f[n], & \text{otherwise} \end{cases} \tag{32}$$

The DFT of the perturbed signal is:

$$F'[k] = \sum_{n=0}^{N-1} f'[n] \cdot e^{-i\frac{2\pi}{N}kn} \tag{33}$$

$$= \sum_{n=0}^{N-1} f[n] \cdot e^{-i\frac{2\pi}{N}kn} + \Delta f \cdot e^{-i\frac{2\pi}{N}kn_0} \tag{34}$$

$$= F[k] + \Delta f \cdot e^{-i\frac{2\pi}{N}kn_0} \tag{35}$$

Thus, the frequency-domain difference is:

$$\Delta F[k] = F'[k] - F[k] = \Delta f \cdot e^{-i\frac{2\pi}{N}kn_0} \tag{36}$$

Since $e^{-i\frac{2\pi}{N}kn_0} \neq 0$ holds for all $k$, we conclude that a perturbation $\Delta f$ at a single time index affects all frequency components, i.e., $\Delta F[k] \neq 0$ for all $k$. This proves the global property of the DFT to changes in the time domain.

### D.2 LOCAL PERTURBATION IN FREQUENCY AFFECTS ALL TIME INDICES

Now consider a perturbation $\Delta f \in \mathbb{C}$ introduced at frequency index $k_0$. The perturbed spectrum is:

$$F'[k] = \begin{cases} F[k] + \Delta f, & k = k_0 \\ F[k], & \text{otherwise} \end{cases} \tag{37}$$

The inverse DFT of the perturbed spectrum becomes:

$$f'[n] = \frac{1}{N} \sum_{k=0}^{N-1} F'[k] \cdot e^{i\frac{2\pi}{N}kn} \tag{38}$$

$$= \frac{1}{N} \sum_{k=0}^{N-1} F[k] \cdot e^{i\frac{2\pi}{N}kn} + \frac{\Delta f}{N} \cdot e^{i\frac{2\pi}{N}k_0 n} \tag{39}$$

$$= f[n] + \frac{\Delta f}{N} \cdot e^{i\frac{2\pi}{N}k_0 n} \tag{40}$$

The corresponding time-domain difference is:

$$\Delta f[n] = f'[n] - f[n] = \frac{\Delta f}{N} \cdot e^{i\frac{2\pi}{N}k_0 n} \tag{41}$$

Since the exponential term remains non-zero for all $n$, it follows that a local perturbation $\Delta f$ at a single frequency index affects all time-domain components, i.e., $\Delta f[n] \neq 0$ for all $n$. This confirms the global property of the inverse DFT to changes in the frequency domain.

### D.3 CONCLUSION

We have rigorously proven that the Discrete Fourier Transform and its inverse are inherently global in nature: any local modification in one domain (time or frequency) induces non-zero changes across all indices in the dual domain. This global property enables SSMs to achieve global receptive fields, precisely regulating the state update and modeling processes of SSMs, thereby resulting in superior performance of SSMs in vision tasks.

## E   PROOF THAT THE 2D-DFT SATISFIES THE PROPOSED DEFINITION

We prove that both the 2D-DFT (forward) and its inverse satisfy the proposed definition.

For an image $\mathbf{X} \in \mathbb{R}^{H \times W}$, the forward and inverse 2D-DFT are defined as follows:

$$\hat{\mathbf{X}}_{u,v} = \sum_{i=0}^{H-1} \sum_{j=0}^{W-1} X_{i,j}\, e^{-2\pi i \left( \frac{ui}{H} + \frac{vj}{W} \right)}, \quad u = 0, \dots, H-1,\ v = 0, \dots, W-1 \tag{42}$$

$$X_{i,j} = \frac{1}{HW} \sum_{u=0}^{H-1} \sum_{v=0}^{W-1} \hat{\mathbf{X}}_{u,v}\, e^{2\pi i \left( \frac{ui}{H} + \frac{vj}{W} \right)}, \quad i = 0, \dots, H-1,\ j = 0, \dots, W-1 \tag{43}$$

### E.1 Forward 2D-DFT Satisfies Global Gradient Dependency

**Function Definition:** For an image $\mathbf{X} \in \mathbb{R}^{H \times W}$, consider the forward transform as the modeling function:

$$f_{forward}(\mathbf{X})_{u,v} = \sum_{i=0}^{H-1} \sum_{j=0}^{W-1} X_{i,j} \cdot e^{-2\pi i \left( \frac{ui}{H} + \frac{vj}{W} \right)}, \quad u = 0, \dots, H-1,\ v = 0, \dots, W-1 \tag{44}$$

**Gradient Computation:** The gradient with respect to input pixel $X_{m,n}$ is:

$$\frac{\partial f_{forward}(\mathbf{X})_{u,v}}{\partial X_{m,n}} = e^{-2\pi i \left( \frac{um}{H} + \frac{vn}{W} \right)} \tag{45}$$

The Frobenius norm over all frequency components:

$$\left\| \frac{\partial f_{forward}(\mathbf{X})}{\partial X_{m,n}} \right\|_F = \sqrt{ \sum_{u=0}^{H-1} \sum_{v=0}^{W-1} \left| e^{-2\pi i \left( \frac{um}{H} + \frac{vn}{W} \right)} \right|^2 } = \sqrt{HW} \tag{46}$$

Define the global influence function:

$$\mathcal{I}_{forward}(m,n) = \sqrt{HW} > 0 \tag{47}$$

Conclusion for Forward Transform:

$$\left\| \frac{\partial f_{forward}(\mathbf{X})}{\partial X_{m,n}} \right\|_F = \sqrt{HW} \geq \mathcal{I}_{forward}(m,n) > 0 \tag{48}$$

### E.2 Inverse 2D-DFT Satisfies Global Gradient Dependency

**Function Definition:** Consider the inverse transform as the modeling function for frequency domain input $\hat{\mathbf{X}} \in \mathbb{C}^{H \times W}$:

$$f_{inverse}(\hat{\mathbf{X}})_{i,j} = \frac{1}{HW} \sum_{u=0}^{H-1} \sum_{v=0}^{W-1} \hat{\mathbf{X}}_{u,v} \cdot e^{2\pi i \left( \frac{ui}{H} + \frac{vj}{W} \right)}, \quad i = 0, \dots, H-1,\ j = 0, \dots, W-1 \tag{49}$$

**Gradient Computation:** The gradient with respect to frequency component $\hat{\mathbf{X}}_{p,q}$ is:

$$\frac{\partial f_{inverse}(\hat{\mathbf{X}})_{i,j}}{\partial \hat{\mathbf{X}}_{p,q}} = \frac{1}{HW} e^{2\pi i \left( \frac{pi}{H} + \frac{qj}{W} \right)} \tag{50}$$

The Frobenius norm over all spatial positions:

$$\left\| \frac{\partial f_{inverse}(\hat{\mathbf{X}})}{\partial \hat{\mathbf{X}}_{p,q}} \right\|_F = \sqrt{\sum_{i=0}^{H-1} \sum_{j=0}^{W-1} \left| \frac{1}{HW} e^{2\pi i \left( \frac{pi}{H} + \frac{qj}{W} \right)} \right|^2} = \frac{1}{\sqrt{HW}} \tag{51}$$

Define the global influence function:

$$\mathcal{I}_{inverse}(p,q) = \frac{1}{\sqrt{HW}} > 0 \tag{52}$$

Conclusion for Inverse Transform:

$$\left\| \frac{\partial f_{inverse}(\hat{\mathbf{X}})}{\partial \hat{\mathbf{X}}_{p,q}} \right\|_F = \frac{1}{\sqrt{HW}} \geq \mathcal{I}_{inverse}(p,q) > 0 \tag{53}$$

### E.3 CONSTRAINT ANALYSIS

The 2D-DFT inherently avoids imposing strict order-dependent constraints, as both the forward and inverse transforms treat all spatial positions and frequency components uniformly, without relying on any specific ordering of the input pixels. As a result, the global gradient dependency property holds regardless of the input arrangement, ensuring that $f$ exhibits the desired order-invariant global modeling behavior.

### E.4 CONCLUSION

Based on the above analysis, we have rigorously demonstrated that both the forward and inverse 2D-DFT satisfy our proposed mathematical definition of global image modeling. Specifically, for any spatial pixel $(i,j)$ or frequency component $(\omega_1, \omega_2)$, the Frobenius norm of the corresponding gradient is strictly positive and bounded below by a constant:

$$\left\| \frac{\partial f(\mathbf{X})}{\partial X_{i,j}} \right\|_F \geq \mathcal{I}(i,j) > 0, \quad \inf_{(i,j)} \mathcal{I}(i,j) \geq \tau > 0, \tag{54}$$

with $\mathcal{I}_{forward} = \sqrt{HW}$ for the forward transform and $\mathcal{I}_{inverse} = 1/\sqrt{HW}$ for the inverse transform.

## F ALGORITHM FLOW OF THE GSSM MODULE

We provide the detailed procedure of GSSM in Algorithm 1. The input feature $\mathbf{X}$ is first linearly mapped into two branches, $\mathbf{x}$ and $\mathbf{x_1}$, and preliminary features are extracted through one-dimensional convolution and SiLU activation. If the frequency domain module is enabled ($use_{freq} = \text{True}$), a 2D DFT is applied to $\mathbf{x}$ to extract high and low frequency components, which are then weighted and reconstructed. The modulation coefficients $\alpha_1$ and $\alpha_2$ are used to guide the fusion of frequency-domain information with the original features. The fused features are then fed into the SSM, where long-range dependencies are modeled via selective scanning. Finally, the result is concatenated with $\mathbf{x_1}$ and projected to obtain the output $\mathbf{X}_{out}$.

## G GMAMBA BLOCK EFFICIENCY ANALYSIS

We follow the computational complexity analysis methodology of Vim (Zhu et al., 2024a), where the input sequence is defined as $L \in \mathbb{R}^{M \times D}$, with $M = H \times W$ denoting the sequence length and $D$ representing the hidden dimension. The computational complexities of Self-Attention (Shaw et al., 2018), Mamba (Gu & Dao, 2023), VMamba (Liu et al., 2024b), and Vim (Zhu et al., 2024a) are given by:

$$\Omega(\text{Self-Attention}) = 4MD^2 + 2M^2D, \quad \Omega(\text{Mamba}) = 8MDN, \tag{55}$$

$$\Omega(\text{Vim}) = 16MDN, \quad \Omega(\text{VMamba}) = 32MDN \tag{56}$$

---

**Algorithm 1** GSSM Module Process

---

**Input**: Hidden states $\mathbf{X} \in \mathbb{R}^{B \times L \times D}$, feature dimension $D$
**Parameter**: Frequency flag $use\_freq$, learnable weights $\{\theta_{low}, \theta_{high}, \theta_{ratio}\}$
**Output**: Global-aware representation $\mathbf{X}_{out} \in \mathbb{R}^{B \times L \times D}$

1: $\mathbf{x}, \mathbf{x_1} \leftarrow \text{Linear}(\mathbf{X})$
2: $\mathbf{x} \leftarrow \text{SiLU}(\text{Conv1d}(\mathbf{x})), \mathbf{x_1} \leftarrow \text{SiLU}(\text{Conv1d}(\mathbf{x_1}))$
3: **if** $use\_freq = \text{True}$ **then**
4:     /* *Frequency Encoding Module* */
5:     $\mathbf{X}_{freq} \leftarrow \text{DFT2D}(\text{reshape}(\mathbf{x}, B, D', H, W))$
6:     $\mathbf{X}_{low}, \mathbf{X}_{high} \leftarrow \text{freq\_decompose}(\mathbf{X}_{freq})$
7:     $\mathbf{F}_{low} \leftarrow \sigma(\theta_{low}) \cdot \text{IDFT2D}(\mathbf{X}_{low})$
8:     $\mathbf{F}_{high} \leftarrow \sigma(\theta_{high}) \cdot \text{IDFT2D}(\mathbf{X}_{high})$
9:     $\mathbf{F}_{global} \leftarrow \text{Concat}([\mathbf{x}, \mathbf{F}_{low}, \mathbf{F}_{high}], \dim = -1)$
10:    /* *Frequency-Guided Modulation* */
11:    $\alpha_1, \alpha_2 \leftarrow \text{Sigmoid}(\text{ConvBlock}(\mathbf{F}_{global}))$
12:    $\mathbf{x} \leftarrow \alpha_1 \cdot \mathbf{x} + \alpha_2 \cdot \mathbf{F}_{global}$
13: **end if**
14: /* *State Space Modeling* */
15: $\mathbf{\Delta t}, \mathbf{B}, \mathbf{C} \leftarrow \text{SSM\_Params}(\mathbf{x})$
16: Discretize: $\mathbf{\Delta t} \leftarrow \text{softplus}(\mathbf{\Delta t}), \mathbf{A} \leftarrow -\exp(\mathbf{A}_{log})$
17: $\mathbf{y} \leftarrow \text{SelectiveScan}(\mathbf{x}, \mathbf{\Delta t}, \mathbf{A}, \mathbf{B}, \mathbf{C})$
18: $\mathbf{X}_{out} \leftarrow \text{Linear}(\text{concat}([\mathbf{y}, \mathbf{x_1}]))$
19: **return** $\mathbf{X}_{out}$

---

where $N$ is the state dimension (typically 16).

In our proposed GMamba, the FEM introduces an additional complexity of $\mathcal{O}(M \log M)$. Therefore, the overall computational complexity of GMamba is:

$$\Omega(\text{GMamba}) = MDN + M \log M. \tag{57}$$

Notably, the complexity of GMamba scales in a **linear-logarithmic** manner with respect to the sequence length $M$, which is significantly more efficient than the quadratic complexity $\mathcal{O}(M^2 D)$ of conventional self-attention mechanisms.

## H  DATASETS AND EXPERIMENTAL DETAILS

### H.1  DETAILED DESCRIPTION OF THE DATASET

**Vaihingen.** This dataset consists of 33 true orthophoto images with 2500×2000 resolution and 9 cm GSD, covering urban areas in Germany. We use only RGB images and exclude DSM data. The dataset includes six semantic categories: impervious surfaces, buildings, low vegetation, trees, cars, and clutter/background. Following standard protocol (Wang et al., 2022), we use 16 images for training and 17 for testing.

**Potsdam.** This dataset contains 38 high-quality tiles with $6000 \times 6000$ resolution and 5 cm GSD. We use only RGB bands and exclude image 7_10 due to annotation errors. The semantic categories are consistent with Vaihingen. Following common practice (Wang et al., 2022), 23 images are used for training and 14 for testing.

**LoveDA.** This large-scale dataset covers urban and rural areas with 5987 images of $1024 \times 1024$ resolution and 0.3 m GSD across three Chinese cities. It includes seven categories: buildings, roads, water, vegetation, vehicles, clutter, and background. The standard split contains 2522/1669/1796 images for train/validation/test.

**UAVid.** This UAV-based video semantic segmentation dataset contains 42 sequences with 4K resolution (3840×2160) captured from low altitudes. It covers diverse urban scenes including residential areas, commercial districts, and industrial zones. The dataset provides annotations for eight semantic classes: building, road, tree, low vegetation, moving car, static car, human, and clutter. Following the official split, we use 20 sequences for training, 7 for validation, and 15 for testing.

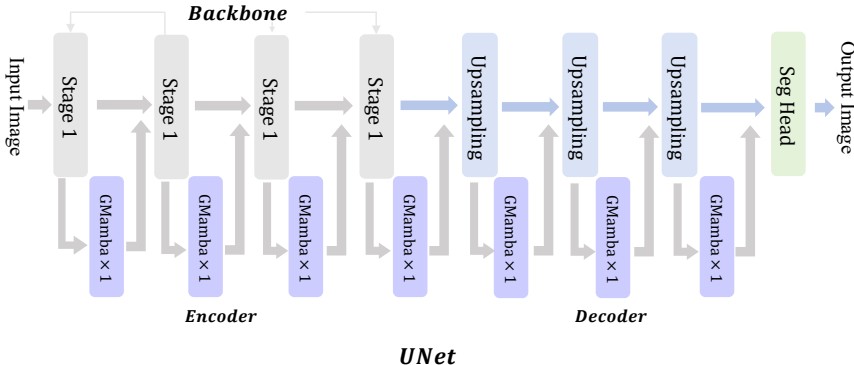

Figure 4: The GMamba module is inserted after each CNN stage to inject global information into the feature representation.

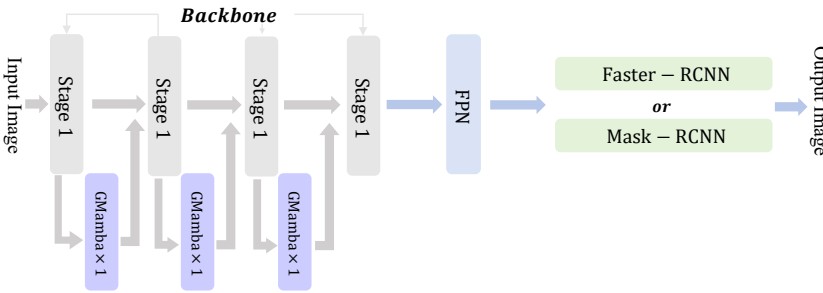

Figure 5: Inserting GMamba modules into the first three stages of the backbone to inject global information into the feature representations.

## H.2 EXPERIMENTAL DETAILS

**Experimental Details on Vaihingen.** We conduct experiments on the Vaihingen dataset using the popular GeoSeg (Wang et al., 2022) framework. To highlight the effectiveness of global modeling, we select the simple UNet network as the baseline and employ pre-trained ResNet34, Swin Transformer-Tiny, and ConvNeXt-Small as UNet backbones. For fair comparison, we adopt GeoSeg's unified training configuration: batch size of 8, weight decay of 2.5e-4, initial learning rate of 6e-4, and training for 100 epochs. As shown in Fig. 4, we insert modules into each output stage of the encoder and decoder via residual connections to provide global information, with a total of 7 modules insertable in the UNet network. All comparison modules (Swin Transformer (Liu et al., 2021), Swin Transformer V2 (Liu et al., 2022), Vim (Zhu et al., 2024a), VMamba (Liu et al., 2024b), and TinyVim (Ma et al., 2024)) are inserted using the same strategy, maintaining identical training configurations, parameter settings, and module counts. Due to the high complexity of the ConvNeXt backbone and Swin Transformer V2, it is not feasible to train the full model on a single RTX 4090 (24GB) GPU. Therefore, we insert only 6 Swin Transformer V2 modules, leaving the last stage unchanged. All experiments are conducted using PyTorch on a single NVIDIA RTX 4090 (24GB) GPU. We follow the official dataset split and cropping strategy, resizing all images to 1024×1024, and report mean Intersection over Union (mIoU), mean F1 score (mF1), and Overall Accuracy (OA) metrics on the test set.

**Experimental details for Potsdam, LoveDA, and UAVid.** Similar to the Vaihingen dataset, we conduct experiments on Potsdam, LoveDA, and UAVid datasets using the GeoSeg framework. The experimental configurations follow GeoSeg's default settings: training for 100 epochs on Potsdam, 30 epochs on LoveDA, and 40 epochs on UAVid, with batch sizes set to 4, 8, and 2, respectively. All datasets are preprocessed using GeoSeg's tool and cropped to 1024×1024 size, with consistent experimental configurations maintained across all comparison modules.

**Experimental details on MS-COCO.** We conduct experiments on the MS-COCO dataset (Lin et al., 2014) using the popular MMDetection (Chen et al., 2019) framework. Pre-trained ResNet50 (He et al., 2016) and Swin Transformer-Tiny (Liu et al., 2021) are adopted as the backbone networks, with the Feature Pyramid Network (FPN) (Lin et al., 2017) serving as the neck architecture. For object detection, we employ Faster R-CNN (Ren et al., 2016) and Mask R-CNN (He et al., 2017) as the detection heads. For instance segmentation, we use Mask R-CNN. To ensure fair comparisons, we follow the default data processing pipeline and hyperparameter configurations provided by MMDetection. Our module is inserted into the first three stages of the backbone in a residual manner to provide global information, as illustrated in Fig. 5. For all comparison modules (Swin Transformer, Swin Transformer V2, VMamba, and TinyVim), we adopt the same data preparation and parameter settings. All experiments are conducted on a server equipped with 8 NVIDIA RTX 4090 GPUs (24GB), with each GPU processing 2 images. The models are trained for 12 epochs following the `schedule_1x` training policy, where the learning rate is decayed by a factor of 10 at the 8th and 11th epochs. We evaluate the average precision (AP) over IoU thresholds ranging from 0.5 to 0.95 with a step size of 0.05. Additionally, we report AP metrics for small, medium, and large objects to provide a comprehensive assessment of model performance.

# I    ADDITIONAL EXPERIMENTS ON SEMANTIC SEGMENTATION

## I.1    COMPLEXITY ANALYSIS

To analyze the efficiency of GMamba, Table 8 demonstrates the comprehensive performance evaluation of GMamba across different backbone architectures on the Vaihingen dataset. Our method consistently outperforms existing state-of-the-art approaches while maintaining computational efficiency. Specifically, GMamba achieves competitive or superior mIoU scores compared to established methods: U-ResNet34 + GMamba reaches 84.74% mIoU, surpassing DeepLabv3+ by 4.01% with significantly fewer parameters (30.96M vs. 59.30M) and reduced computational cost (36.30G vs. 181.20G FLOPs). When integrated with more powerful backbones, GMamba demonstrates scalable performance gains, with U-ConvNeXt(S) + GMamba achieving the highest mIoU of 86.00%, outperforming recent methods like XNet and SparseFormer. The consistent improvements across all three backbone architectures—ResNet34 (+3.09%), Swin Transformer-Tiny (+2.39%), and ConvNeXt-Small (+2.89%)—validate GMamba's universal applicability and plug-and-play nature. While GMamba introduces moderate computational overhead (approximately 22% increase in parameters and 20% in FLOPs), this trade-off is justified by substantial accuracy gains, demonstrating an effective balance between performance and efficiency for practical deployment scenarios.

Table 8: Performance comparison of semantic segmentation methods on Vaihingen dataset and comprehensive ablation study of GMamba enhancement. Evaluation metrics include parameters (M), FLOPs (G), model size (MB), and mean Intersection over Union (mIoU).

| Category | Model | Params (M) | FLOPs (G) | Model Size (MB) | mIoU (%) |
|---|---|---|---|---|---|
| Existing Methods | DeepLabv3+ (Chen et al., 2018) | 59.30 | 181.20 | 226.20 | 80.73 |
| | FT-UNetFormer (Wang et al., 2022) | 96.00 | 128.40 | 366.30 | 84.10 |
| | CGGLNet (Ni et al., 2024) | 34.83 | 605.00 | 132.90 | 83.89 |
| | SparseFormer (Chen et al., 2024c) | 42.52 | 41.95 | 162.17 | 84.12 |
| | XNet (Zhou et al., 2023) | 41.42 | 115.09 | 158.03 | 84.43 |
| Baseline Backbones | UNet-ResNet34 | 25.33 | 30.32 | 96.85 | 81.65 |
| | UNet-Swin(T) | 36.48 | 44.46 | 139.56 | 82.44 |
| | UNet-ConvNeXt(S) | 58.42 | 68.88 | 223.12 | 83.11 |
| + GMamba (Ours) | UNet-ResNet34 + GMamba | **30.96** | **36.30** | **118.45** | **84.74** |
| | UNet-Swin(T) + GMamba | **49.13** | **57.81** | **187.89** | **84.83** |
| | UNet-ConvNeXt(S) + GMamba | **71.06** | **85.66** | **271.39** | **86.00** |

## I.2    ABLATION STUDY ON GMAMBA ENCODER-DECODER INTEGRATION

To validate the effectiveness of GMamba modules in providing global information at different network stages, we conduct a detailed analysis of GMamba insertion positions and quantities across

different backbone networks on the Vaihingen dataset, with results shown in Tables 9, 10, and 11. Integrating GMamba in both encoder and decoder stages achieves optimal performance improvements across all three backbone networks (U-ConVNeXt-Small, U-Swin(T), and U-ResNet34), with mIoU improvements of +2.89%, +2.39%, and +3.09%, respectively. The analysis reveals that GMamba integration in the encoder typically provides more significant performance gains than decoder-only integration, particularly evident in U-ConVNeXt-Small (+1.59% vs +1.02% mIoU) and U-ResNet34 (+1.73% vs +1.59% mIoU), indicating that early-stage global context modeling in the encoder plays a crucial role in effective feature representation. Notably, the joint encoder-decoder integration demonstrates significant synergistic effects, with performance substantially exceeding the simple summation of individual contributions, proving GMamba's complementary roles in global context capture and feature refinement. From a computational efficiency perspective, all architectures maintain reasonable computational overhead, with parameter increases of 22%-35% and FLOP increases of 17%-30%, while achieving accuracy improvements exceeding 2% mIoU.

Table 9: Ablation study of GMamba integration in Encoder and Decoder based on U-ConVNext-Small.

| Model Variant | Encoder | Decoder | Params (M) | FLOPs (G) | mIoU (%) | F1 (%) | OA (%) |
|---|---|---|---|---|---|---|---|
| Baseline (UNet-ConVNext-Small) | – | – | 58.42 | 68.88 | 83.11 | 90.19 | 92.30 |
| + GMamba in Encoder | ✓ × 4 | – | 68.55 | 76.87 | 84.70 | 91.52 | 93.59 |
| + GMamba in Decoder | – | ✓ × 3 | 60.93 | 74.24 | 84.13 | 91.16 | 93.35 |
| + GMamba in Both | ✓ × 4 | ✓ × 3 | 71.06 | 85.66 | 86.00 | 92.31 | 93.99 |

Table 10: Ablation study of GMamba integration in Encoder and Decoder based on U-Swin(T).

| Model Variant | Encoder | Decoder | Params (M) | FLOPs (G) | mIoU (%) | F1 (%) | OA (%) |
|---|---|---|---|---|---|---|---|
| Baseline (UNet-Swin(T)) | – | – | 36.48 | 44.46 | 82.44 | 89.75 | 92.18 |
| + GMamba in Encoder | ✓ × 4 | – | 46.62 | 52.45 | 83.26 | 90.64 | 93.12 |
| + GMamba in Decoder | – | ✓ × 3 | 39.00 | 49.82 | 83.46 | 90.76 | 93.07 |
| + GMamba in Both | ✓ × 4 | ✓ × 3 | 49.13 | 57.81 | 84.83 | 91.61 | 93.65 |

Table 11: Ablation study of GMamba integration in Encoder and Decoder based on U-ResNet34.

| Model Variant | Encoder | Decoder | Params (M) | FLOPs (G) | mIoU (%) | F1 (%) | OA (%) |
|---|---|---|---|---|---|---|---|
| Baseline (UNet-ResNet34) | – | – | 25.33 | 30.32 | 81.65 | 89.24 | 91.86 |
| + GMamba in Encoder | ✓ × 4 | – | 29.84 | 33.89 | 83.38 | 90.72 | 93.12 |
| + GMamba in Decoder | – | ✓ × 3 | 26.45 | 32.73 | 83.24 | 90.63 | 93.13 |
| + GMamba in Both | ✓ × 4 | ✓ × 3 | 30.96 | 36.30 | 84.74 | 91.56 | 93.72 |

## I.3 VISUALIZATION AND ANALYSIS OF SEMANTIC SEGMENTATION RESULTS

We further visualized the segmentation results on four remote sensing datasets before and after integrating GMamba into different backbone networks, as shown in Figs. 6–9. The black boxes highlight the performance differences between models with and without GMamba. The results demonstrate that GMamba integration can consistently improve segmentation performance across datasets, exhibiting remarkable generalization capabilities and global modeling advantages.

As illustrated in Fig. 6 and Fig. 7, on the Vaihingen and Potsdam datasets, the incorporation of GMamba significantly enhances the segmentation performance of complex-shaped objects such as buildings and vehicles, which typically exhibit spatial distribution characteristics throughout the entire scene. This performance improvement fully demonstrates that GMamba can effectively integrate information from distant regions, thereby overcoming the inherent limitations of local receptive fields. Similarly, in the LoveDA dataset experiments shown in Fig. 8, large-scale categories such as buildings and agricultural land achieve higher segmentation accuracy, reflecting GMamba's

capability to maintain object boundary consistency and reduce misclassification rates in complex and heterogeneous geographical landscapes. The results in Fig. 9 show that GMamba enhances the model's ability to extract global information on the UAVid dataset, enabling more accurate identification of spatially extended objects such as roads, rivers, and large buildings, which typically span multiple local patches. These observations collectively validate the effectiveness of GMamba in global feature modeling, as well as its superior generalization performance across different datasets and backbone architectures.

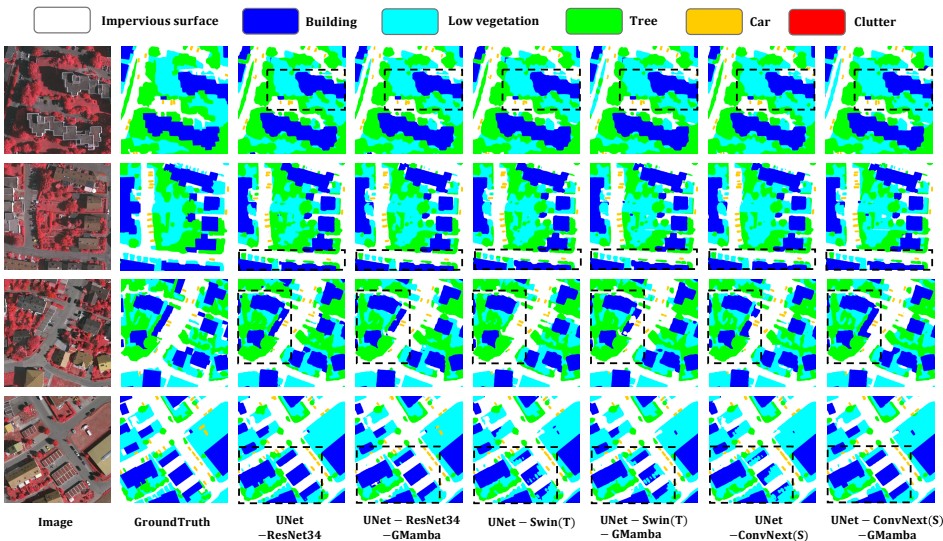

Figure 6: Segmentation results on the Vaihingen dataset with different backbones, before and after integrating GMamba. Black boxes highlight improvements in globally-dependent classes such as Buildings and Cars.

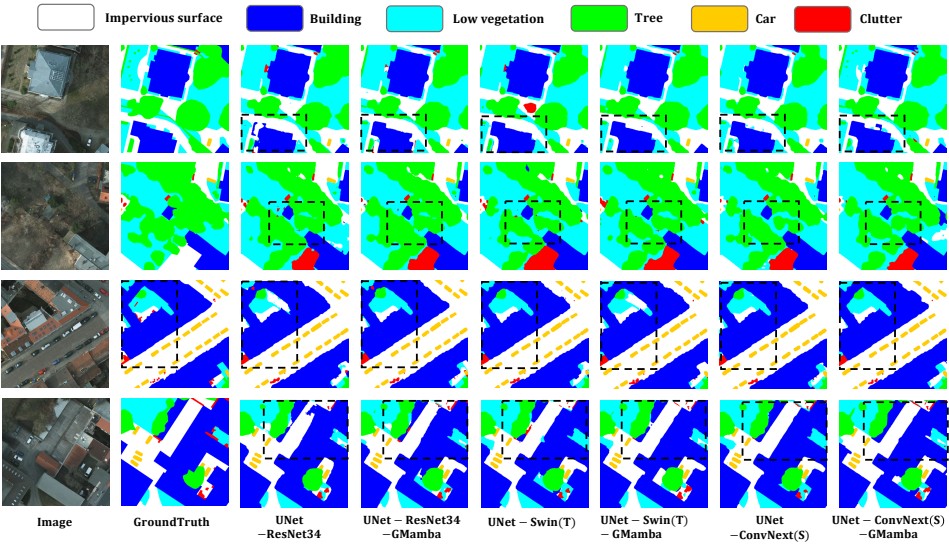

Figure 7: Segmentation results on the Potsdam dataset with different backbones, before and after integrating GMamba. GMamba enhances global context modeling for classes like Buildings and Cars.

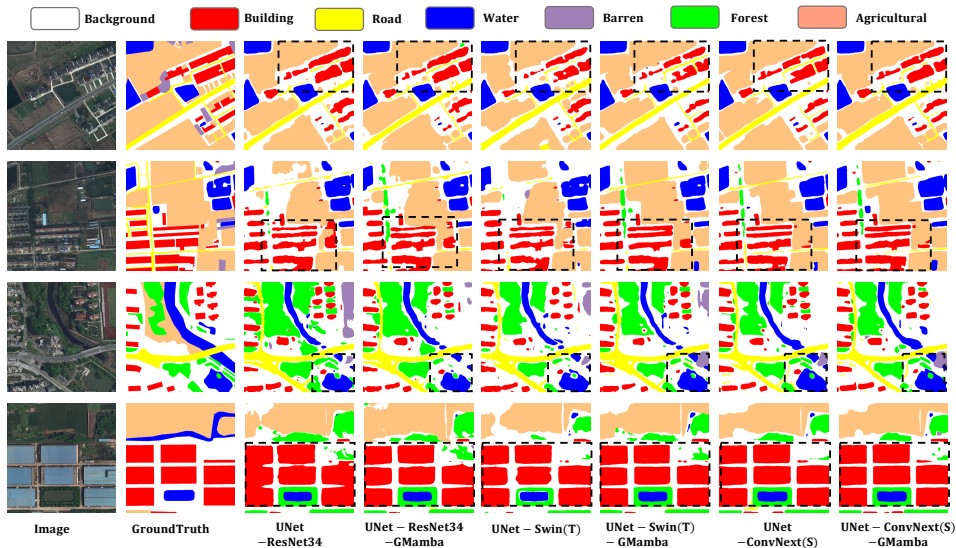

Figure 8: Segmentation results on the LoveDA dataset with different backbones, before and after integrating GMamba. GMamba improves segmentation of large-scale classes such as Buildings and Agriculture.

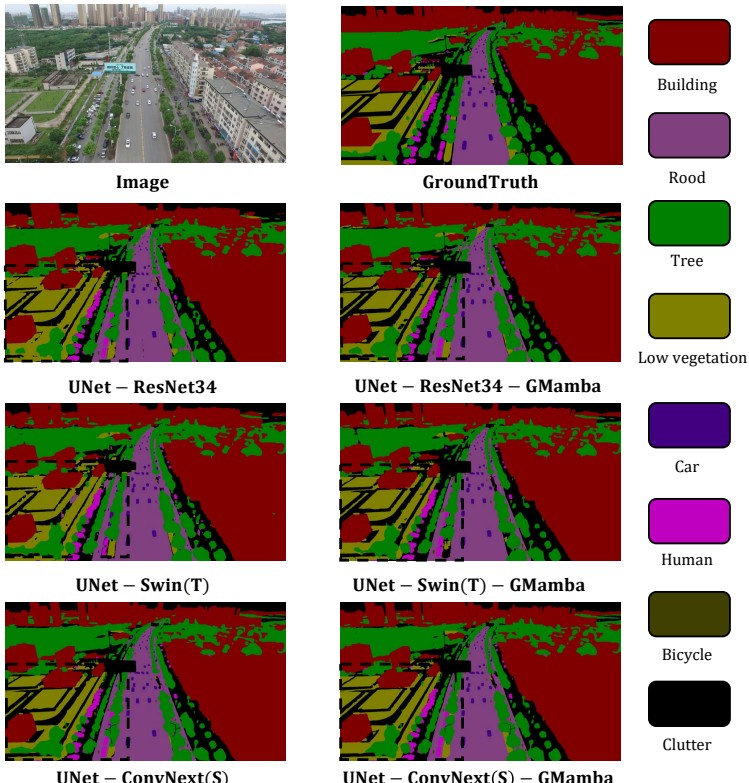

Figure 9: Segmentation results on the UAVid dataset with different backbones, before and after integrating GMamba. GMamba strengthens global information extraction, enabling better recognition of spatially extended objects.

## J ADDITIONAL EXPERIMENTS ON OBJECT DETECTION AND INSTANCE SEGMENTATION

### J.1 COMPLETE COMPARISON EXPERIMENTS ON OBJECT DETECTION

In Table 12, we present a complete comparison between GMamba and other global modeling modules on the MS-COCO dataset for object detection, across different backbones (ResNet-50 and Swin-T) as well as model architectures.

Table 12: Comparison of GMamba and other global modeling modules inserted into the first three stages of different backbones (ResNet-50 / Swin-T) on the MS-COCO object detection task with Faster R-CNN and Mask R-CNN.

| Model Variant | AP (%) | AP$_{50}$ (%) | AP$_{75}$ (%) | AP$_S$ (%) | AP$_M$ (%) | AP$_L$ (%) | Params (M) | FLOPs (G) |
|---|---|---|---|---|---|---|---|---|
| *Faster R-CNN* | | | | | | | | |
| Baseline (ResNet50) | 37.2 | 57.8 | 40.4 | 21.5 | 40.6 | 48.0 | 43.80(23.5) | 207.07(76.50) |
| + Swin (×3) | 38.0 | 59.1 | 41.0 | **23.0** | 41.5 | 48.4 | 76.70(56.4) | 210.57(80.00) |
| + SwinV2 (×3) | 37.7 | 58.7 | 40.9 | 21.5 | 41.3 | 48.8 | 76.90(56.6) | 210.37(79.80) |
| + TinyViM (×3) | 37.6 | 58.6 | 40.6 | 21.6 | 41.3 | 48.4 | 61.40(41.1) | 208.87(78.30) |
| + VMamba (×3) | 37.6 | 58.8 | 40.8 | 21.5 | 41.4 | 48.8 | 65.00(44.7) | 209.67(79.10) |
| + Mamba Version (×3) | 37.7 | 58.9 | 41.0 | 21.6 | 41.4 | 48.5 | 65.40(45.1) | 209.97(79.40) |
| + Spatial Mamba (×3) | 37.5 | 58.6 | 40.8 | 21.4 | 41.2 | 48.3 | 64.30(44.0) | 209.27(78.70) |
| + FreqMamba (×3) | 37.6 | 58.7 | 40.9 | 21.5 | 41.3 | 48.4 | 64.90(44.6) | 209.57(79.00) |
| + Group Mamba (×3) | 37.5 | 58.5 | 40.5 | 21.3 | 41.0 | 48.5 | 64.80(44.5) | 209.37(78.80) |
| + GMamba (×3) (Ours) | **38.5** | **59.6** | **42.2** | 22.1 | **42.0** | **49.9** | 61.40(41.1) | 210.22(79.65) |
| Baseline (Swin(T)) | 41.6 | 64.0 | 45.2 | 25.7 | 44.8 | 55.4 | 45.15(27.5) | 213.00(86.10) |
| + Swin (×3) | 42.6 | 64.8 | 46.7 | 26.5 | 45.9 | 56.1 | 63.85(46.2) | 216.50(89.60) |
| + SwinV2 (×3) | 42.4 | 65.1 | 46.3 | 25.8 | 45.9 | 55.9 | 63.95(46.3) | 216.30(89.40) |
| + TinyViM (×3) | 41.9 | 64.7 | 45.8 | 26.7 | 45.4 | 54.3 | 55.45(37.8) | 214.80(87.90) |
| + VMamba (×3) | 42.2 | 64.7 | 46.2 | 26.7 | 45.4 | 55.1 | 57.95(40.3) | 215.60(88.70) |
| + Mamba Version (×3) | 42.3 | 64.5 | 46.4 | 26.0 | 45.5 | 55.5 | 63.75(46.1) | 216.50(89.60) |
| + Spatial Mamba (×3) | 42.0 | 64.2 | 46.1 | 25.8 | 45.2 | 55.2 | 63.55(45.9) | 216.20(89.30) |
| + FreqMamba (×3) | 42.1 | 64.3 | 46.2 | 25.9 | 45.3 | 55.3 | 63.65(46.0) | 216.30(89.40) |
| + Group Mamba (×3) | 42.4 | 65.1 | 46.4 | 25.8 | 45.7 | 55.4 | 61.15(43.5) | 215.90(89.00) |
| + GMamba (×3) (Ours) | **42.9** | **65.6** | **46.9** | **27.1** | **46.4** | **56.4** | 55.25(37.6) | 216.15(89.25) |
| *Mask R-CNN* | | | | | | | | |
| Baseline (ResNet50) | 37.5 | 58.1 | 41.3 | 21.2 | 40.9 | 48.3 | 46.45(23.5) | 260.14(76.50) |
| + Swin (×3) | 38.4 | 59.2 | 41.7 | 21.6 | 42.2 | 49.3 | 79.35(56.4) | 263.64(80.00) |
| + SwinV2 (×3) | 38.3 | 59.1 | 41.7 | 22.2 | 41.9 | 49.6 | 79.55(56.6) | 263.44(79.80) |
| + TinyViM (×3) | 38.2 | 58.8 | 41.6 | 21.5 | 41.6 | 49.9 | 64.05(41.1) | 261.94(78.30) |
| + VMamba (×3) | 38.4 | 59.1 | 41.9 | 22.3 | 41.8 | 49.8 | 67.65(44.7) | 262.74(79.10) |
| + Mamba Version (×3) | 38.2 | 59.8 | 42.2 | 22.3 | 42.0 | 49.2 | 68.05(45.1) | 263.04(79.40) |
| + Spatial Mamba (×3) | 38.0 | 59.4 | 42.0 | 22.0 | 41.8 | 49.0 | 66.95(44.0) | 262.34(78.70) |
| + FreqMamba (×3) | 38.1 | 59.6 | 42.1 | 22.1 | 41.9 | 49.1 | 67.55(44.6) | 262.64(79.00) |
| + Group Mamba (×3) | 38.3 | 59.0 | 41.8 | 22.6 | 41.4 | 50.2 | 67.45(44.5) | 262.44(78.80) |
| + GMamba (×3) (Ours) | **39.1** | **60.1** | **42.8** | **23.3** | **42.5** | **50.3** | 64.05(41.1) | 263.29(79.65) |
| Baseline (Swin(T)) | 42.2 | 64.7 | 46.0 | 26.7 | 45.5 | 55.6 | 47.79(27.5) | 266.00(86.10) |
| + Swin (×3) | 43.1 | 65.3 | 47.1 | 27.4 | 46.4 | 56.4 | 66.49(46.2) | 269.50(89.60) |
| + SwinV2 (×3) | 43.3 | 65.4 | 47.8 | **28.2** | 47.0 | **57.4** | 67.49(46.3) | 269.30(89.40) |
| + TinyViM (×3) | 42.7 | 65.1 | 46.6 | 26.8 | 46.0 | 56.3 | 58.09(37.8) | 267.80(87.90) |
| + VMamba (×3) | 43.0 | 65.3 | 47.3 | 27.2 | 46.5 | 56.6 | 60.59(40.3) | 268.60(88.70) |
| + Mamba Version (×3) | 43.0 | 65.1 | 47.5 | 27.3 | 46.7 | 57.0 | 66.39(46.1) | 269.50(89.60) |
| + Spatial Mamba (×3) | 42.8 | 64.9 | 47.2 | 27.1 | 46.5 | 56.8 | 66.19(45.9) | 269.20(89.30) |
| + FreqMamba (×3) | 42.9 | 65.0 | 47.3 | 27.2 | 46.6 | 56.9 | 66.29(46.0) | 269.30(89.40) |
| + Group Mamba (×3) | 43.0 | 65.2 | 47.0 | 27.3 | 46.6 | 56.5 | 63.79(43.5) | 268.90(89.00) |
| + GMamba (×3) (Ours) | **43.7** | **66.4** | **48.0** | 27.5 | **47.3** | 57.0 | 57.89(37.6) | 269.15(89.25) |

### J.2 ABLATION STUDY ON INSERTING GMAMBA INTO DIFFERENT STAGES

To evaluate the effectiveness of GMamba integration at different stages of the backbone, we conduct ablation studies on both object detection and instance segmentation tasks using Faster R-CNN and Mask R-CNN with ResNet50 and Swin(T) backbones on the MS-COCO dataset. Tables 13 and 15 show the results for Faster R-CNN, while Tables 14, 16, and 17 present the results for Mask R-CNN for object detection and instance segmentation. Inserting GMamba modules progressively from Stage 1 to Stage 3 consistently improves performance across all backbones and frameworks. For example, in Faster R-CNN with ResNet50 (Table 13), AP increases from 37.2% (baseline) to 38.5% when GMamba is applied to all stages, and in Mask R-CNN with Swin(T) for instance segmentation (Table 17), AP improves from 38.7% to 39.8%. Stage-wise analysis indicates that early-stage enhancement provides modest gains, while late-stage integration contributes most significantly, highlighting the benefit of enhancing high-level semantic features. The computational overhead introduced by GMamba is moderate, with parameters and FLOPs increasing reasonably relative to the performance gains. Consistent improvements across backbone types and tasks demonstrate the robustness and universality of GMamba.

Table 13: Ablation study on inserting GMamba modules into different stages (Stage 1/2/3) of the ResNet50 backbone in Faster R-CNN for object detection on the MS-COCO dataset.

| Inserted Stages | AP (%) | $AP_{50}$ (%) | $AP_{75}$ (%) | $AP_S$ (%) | $AP_M$ (%) | $AP_L$ (%) | Params (M) | FLOPs (G) |
|---|---|---|---|---|---|---|---|---|
| Baseline (No GMamba) | 37.2 | 57.8 | 40.4 | 21.5 | 40.6 | 48.0 | 43.8(23.5) | 207.07(76.50) |
| + GMamba ×1 (Stage 1) | 37.5 | 58.7 | 40.9 | 22.1 | 41.1 | 48.1 | 44.7(24.4) | 207.22(76.65) |
| + GMamba ×2 (Stages 1–2) | 37.9 | 58.7 | 41.3 | 21.4 | 41.4 | 49.8 | 48.0(27.7) | 207.83(77.26) |
| + GMamba ×3 (Stages 1–3) | **38.5** | **59.6** | **42.2** | **22.1** | **42.0** | **49.9** | 61.4(41.1) | 210.22(79.65) |

Table 14: Ablation study on inserting GMamba modules into different stages (Stage 1/2/3) of the ResNet50 backbone in Mask R-CNN for object detection on the MS-COCO dataset.

| Inserted Stages | AP (%) | $AP_{50}$ (%) | $AP_{75}$ (%) | $AP_S$ (%) | $AP_M$ (%) | $AP_L$ (%) | Params (M) | FLOPs (G) |
|---|---|---|---|---|---|---|---|---|
| Baseline (No GMamba) | 37.5 | 58.1 | 41.3 | 21.2 | 40.9 | 48.3 | 46.45(23.5) | 260.14(76.50) |
| + GMamba ×1 (Stage 1) | 37.7 | 58.3 | 40.9 | 21.5 | 41.0 | 48.7 | 47.35(24.4) | 260.29(76.65) |
| + GMamba ×2 (Stages 1–2) | 38.1 | 59.2 | 41.3 | 21.6 | 41.4 | 49.7 | 50.65(27.7) | 260.90(77.26) |
| + GMamba ×3 (Stages 1–3) | **39.1** | **60.1** | **42.8** | **23.3** | **42.5** | **50.3** | 64.05(41.1) | 263.29(79.65) |

Table 15: Ablation study on inserting GMamba modules into different stages (Stage 1/2/3) of the Swin(T) backbone in Faster R-CNN for object detection on the MS-COCO dataset.

| Inserted Stages | AP (%) | $AP_{50}$ (%) | $AP_{75}$ (%) | $AP_S$ (%) | $AP_M$ (%) | $AP_L$ (%) | Params (M) | FLOPs (G) |
|---|---|---|---|---|---|---|---|---|
| Baseline (No GMamba) | 41.6 | 64.0 | 45.2 | 25.7 | 44.8 | 55.4 | 45.15(27.5) | 213.00(86.10) |
| + GMamba ×1 (Stage 1) | 41.9 | 64.7 | 45.7 | 27.0 | 45.3 | 54.9 | 45.65(28.0) | 213.15(86.25) |
| + GMamba ×2 (Stages 1–2) | 42.2 | 64.8 | 45.7 | 26.5 | 45.5 | 55.7 | 47.65(30.0) | 213.76(86.86) |
| + GMamba ×3 (Stages 1–3) | **42.9** | **65.6** | **46.9** | **27.1** | **46.4** | **56.4** | 55.25(37.6) | 216.15(89.25) |

Table 16: Ablation study on inserting GMamba modules into different stages (Stage 1/2/3) of the Swin(T) backbone in Mask R-CNN for object detection on the MS-COCO dataset.

| Inserted Stages | AP (%) | $AP_{50}$ (%) | $AP_{75}$ (%) | $AP_S$ (%) | $AP_M$ (%) | $AP_L$ (%) | Params (M) | FLOPs (G) |
|---|---|---|---|---|---|---|---|---|
| Baseline (No GMamba) | 42.2 | 64.7 | 46.0 | 26.7 | 45.5 | 55.6 | 47.79(27.5) | 266.00(86.10) |
| + GMamba ×1 (Stage 1) | 42.6 | 65.0 | 46.6 | 26.4 | 45.8 | 56.6 | 48.29(28.0) | 266.15(86.25) |
| + GMamba ×2 (Stages 1–2) | 42.9 | 65.5 | 46.8 | 26.5 | 46.1 | 56.8 | 50.29(30.0) | 266.76(86.86) |
| + GMamba ×3 (Stages 1–3) | **43.7** | **66.4** | **48.0** | **27.5** | **47.3** | **57.0** | 57.89(37.6) | 269.15(89.25) |

Table 17: Ablation study on inserting GMamba modules into different stages (Stage 1/2/3) of the Swin(T) backbone in Mask R-CNN for instance segmentation on the MS-COCO dataset.

| Inserted Stages | AP (%) | $AP_{50}$ (%) | $AP_{75}$ (%) | $AP_S$ (%) | $AP_M$ (%) | $AP_L$ (%) | Params (M) | FLOPs (G) |
|---|---|---|---|---|---|---|---|---|
| Baseline (No GMamba) | 38.7 | 61.3 | 41.5 | **20.2** | 41.6 | 56.7 | 47.79(27.5) | 266.00(86.10) |
| + GMamba ×1 (Stage 1) | 39.0 | 61.8 | 42.2 | 19.7 | 41.9 | 57.0 | 48.29(28.0) | 266.15(86.25) |
| + GMamba ×2 (Stages 1–2) | 39.5 | 62.2 | 42.7 | 20.0 | 42.4 | 57.7 | 50.29(30.0) | 266.76(86.86) |
| + GMamba ×3 (Stages 1–3) | **39.8** | **62.7** | **42.8** | 20.1 | **42.7** | **58.0** | 57.89(37.6) | 269.15(89.25) |

## J.3 VISUALIZATION AND ANALYSIS OF OBJECT DETECTION AND INSTANCE SEGMENTATION RESULTS

As shown in Fig. 10, we visualize the object detection and instance segmentation results of Mask R-CNN and Faster R-CNN on the MS-COCO dataset with different backbone networks, enhanced by our GMamba module. It can be observed that for large objects requiring global contextual information, the integration of GMamba significantly improves segmentation accuracy, enabling more precise delineation of object boundaries. Moreover, GMamba consistently maintains high segmentation performance across different object categories, demonstrating its strong global modeling capability as well as its plug-and-play versatility and efficiency.

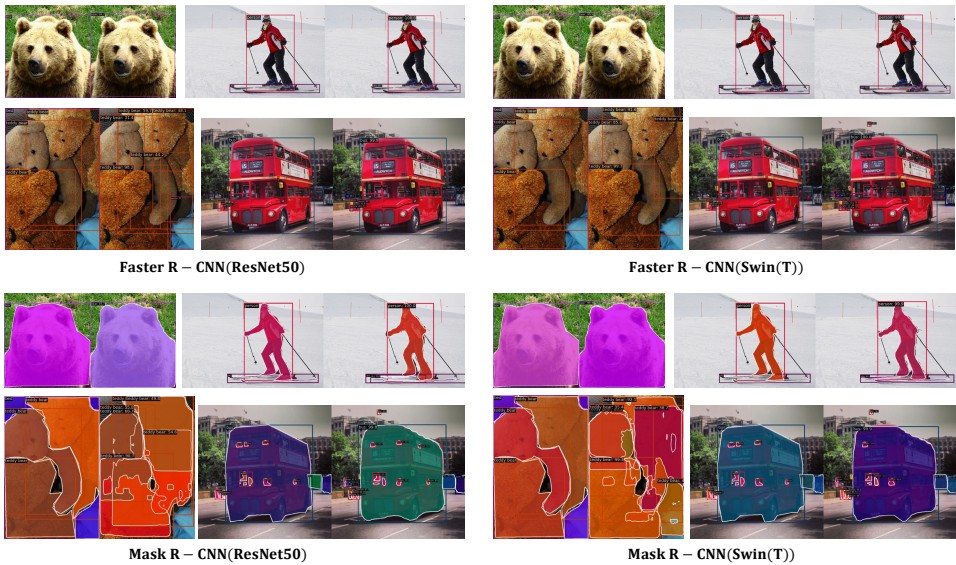

Figure 10: Visualization of object detection and instance segmentation results of Mask R-CNN and Faster R-CNN on the MS-COCO dataset with different backbones.

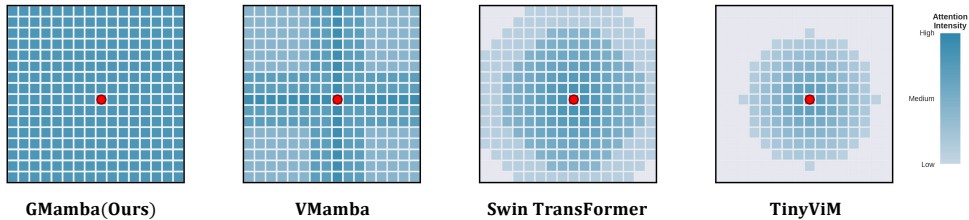

Figure 11: Receptive field distributions of GMamba, Swin Transformer, VMamba, and TinyViM.

# K  ADDITIONAL ABLATION STUDIES ON THE GSSM MODULE

## K.1  COMPARISON OF EXISTING GLOBAL MODELING METHODS

As shown in Table 18, existing heuristic global modeling methods, such as Self-Attention, Vanilla SSM, and VMamba, either rely on token-wise attention or recursive/multi-directional state propagation. While they provide certain levels of globality, they generally suffer from limited gradient sensitivity, weak positional consistency, and lack rigorous theoretical guarantees. In contrast, our proposed GSSM leverages DFT-guided state updates, which integrate frequency-domain global information into the state modeling process. This design ensures gradient sensitivity, preserves positional consistency, and is supported by strong theoretical foundations, distinguishing our approach from heuristic methods and providing a rigorous mathematical foundation for image global modeling.

Table 18: Comparison of Global Modeling Methods with Respect to Theoretical Properties

| Method | Globality | Gradient Dependence | Positional Consistency | Theoretical Guarantee |
|---|---|---|---|---|
| Self-Attention | Full token-wise interaction | Partially satisfied (depends on attention weights) | Yes | Weak |
| Vanilla SSM | Recursive accumulation | No (exponential decay) | No (depends on sequence order) | Weak |
| VMamba | Multi-directional scanning | No (still decays) | No | Weak |
| DFT | Frequency-domain transform | Yes | Yes | Strong |
| GSSM (Ours) | DFT-guided SSM | Yes | Yes | Strong |

## K.2 ANALYSIS OF RECEPTIVE FIELD CHARACTERISTICS

To compare the receptive field sizes of GMamba with other global modeling modules, we visualize the receptive field distributions of different modules in Fig. 11. GMamba, leveraging the DFT-based global modulation mechanism, achieves true global perception with a uniform distribution. VMamba also realizes global perception through its four-directional scanning mechanism, but the sequential nature results in a cross-shaped distribution and limits its practical applicability in image tasks. Swin Transformer achieves global perception as well, but exhibits higher attention in the center and lower attention toward the edges. TinyViM, as a lightweight global modeling module, has a smaller receptive field coverage.

## K.3 EFFECT OF FREQUENCY-DOMAIN INFORMATION

Table 19: Ablation Study on Frequency-domain Information

| Model Variant | mIoU (%) | mF1 (%) | OA (%) |
|---|---|---|---|
| w/o Frequency (NoFreq) | 84.01 | 91.01 | 93.38 |
| + High-Frequency Only (HF Only) | 85.20 | 91.83 | 93.74 |
| + Low-Frequency Only (LF Only) | 85.36 | 91.94 | 93.78 |
| + HF + LF (Simple Addition) | 85.53 | 92.03 | 93.89 |
| + HF + LF (Adaptive Modulation, Ours) | **86.00** | **92.31** | **93.99** |

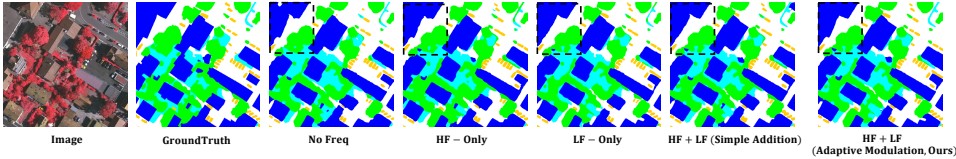

Figure 12: Visualization of ablation experiments on frequency-domain information.

To validate the effectiveness of frequency decomposition and adaptive modulation in GSSM, we conduct a detailed ablation study to investigate the contributions of different frequency components. As shown in Table 19, removing frequency-domain information (NoFreq) leads to a noticeable performance drop (84.01% mIoU), suggesting that purely spatial modeling is insufficient to capture comprehensive semantic cues. When incorporating single-frequency components, low-frequency information (85.36% mIoU) slightly outperforms high-frequency information (85.20% mIoU), which confirms that low-frequency components preserve richer global semantics and contextual structures that are critical for visual understanding. Conversely, high-frequency components, though slightly weaker in isolation, contribute to fine-grained boundary delineation and local details, which remain indispensable for precise predictions.

When both high- and low-frequency signals are combined, the model with simple addition achieves 85.53% mIoU, indicating that complementary interactions exist between global semantics and fine details. However, naive summation still underutilizes their distinct contributions, as it treats all frequency signals with equal importance regardless of spatial context. Our adaptive modulation strategy further improves performance to 86.00% mIoU and 92.31% mF1, achieving a 0.47pp gain over simple addition. This demonstrates the necessity of dynamically learning frequency-domain weights, allowing the model to selectively emphasize semantic-rich low-frequency cues while adaptively integrating high-frequency details depending on scene complexity. These results validate both the rationality of frequency decomposition and the effectiveness of our proposed adaptive modulation, highlighting its role in achieving a better balance between global consistency and local precision.

As shown in Fig. 12, the visualization of different frequency components further validates our observations. Without frequency information (NoFreq), the model struggles to capture global semantic structures, resulting in blurred boundaries and missing regions. Incorporating low-frequency components improves the consistency of large semantic regions, but local details remain underrepresented.

In contrast, high-frequency components enhance edge sharpness and fine-grained details, yet often introduce internal noise. Simply summing low- and high-frequency signals partially mitigates these issues, but still fails to fully exploit their complementary strengths. Our adaptive modulation effectively integrates both global semantics and local details, producing predictions that are both spatially coherent and boundary-precise, demonstrating the necessity of dynamically learning frequency-domain weights.

### K.4 ROBUSTNESS AND CROSS-DOMAIN GENERALIZATION ANALYSIS

To further evaluate the robustness and cross-domain generalization capability of the GMamba model under complex environments, we adopt UNet-ConvNeXt (S) as the baseline and apply typical input perturbations including Gaussian noise (0.01), Gaussian blur (k=3), and partial occlusion (5%). As shown in Table 20, GMamba consistently achieves the best performance under all perturbation settings, with a 5.23% improvement under Gaussian noise and a 5.35% gain under blur, highlighting its strong noise resistance and stability. In addition, to assess the model's adaptability to cross-domain scenarios, we conduct domain generalization experiments on the LoveDA dataset. As reported in Table 21, GMamba not only enhances semantic representation but also significantly alleviates domain shift, maintaining superior and stable segmentation performance across different domains.

Table 20: Robustness Evaluation under Typical Input Perturbations

| Method | Clean (%) | Noise (0.01) (%) | Blur (k=3) (%) | Occlusion (5%) (%) |
|---|---|---|---|---|
| UNet-ConvNext(S) | 83.11 | 77.55 | 79.15 | 77.00 |
| +Swin | 84.82 | 79.10 | 80.30 | 78.20 |
| +SwinV2 | 84.36 | 79.25 | 80.45 | 78.35 |
| +ViM | 84.24 | 79.00 | 80.20 | 78.10 |
| +VMamba | 84.56 | 79.40 | 80.60 | 78.50 |
| +TinyViM | 84.38 | 79.35 | 80.55 | 78.40 |
| +Mamba Version | 84.80 | 79.70 | 81.10 | 78.90 |
| +Spatial Mamba | 84.50 | 79.45 | 80.85 | 78.60 |
| +FreqMamba | 84.60 | 80.10 | 81.50 | 79.20 |
| **+GMamba (Ours)** | **86.00** | **82.78** | **84.50** | **81.00** |

Table 21: Cross-Domain Evaluation on LoveDA Dataset

| Method | Urban → Rural | | Rural → Urban | |
|---|---|---|---|---|
| | mIoU (%) | mF1-score (%) | mIoU (%) | mF1-score (%) |
| UNet-ConVNext(S) | 39.13 | 53.48 | 52.15 | 68.12 |
| +Swin | 41.05 | 55.25 | 54.10 | 70.05 |
| +SwinV2 | 41.18 | 55.40 | 54.22 | 70.12 |
| +TinyViM | 40.50 | 54.90 | 53.70 | 69.80 |
| +ViM | 40.20 | 54.70 | 53.45 | 69.50 |
| +VMamba | 39.80 | 54.30 | 53.10 | 69.10 |
| +Mamba Version | 40.00 | 54.50 | 53.30 | 69.30 |
| +Spatial Mamba | 40.40 | 54.85 | 53.55 | 69.60 |
| +FreqMamba | 41.00 | 55.10 | 54.00 | 69.95 |
| **+GMamba (Ours)** | **42.89** | **58.38** | **55.63** | **71.00** |

### K.5 FAIR COMPARISON WITH PARAMETER-MATCHED BASELINES

To further verify the effectiveness of GMamba under fair comparison conditions, we adjust the channel widths of different backbone networks so that their parameter sizes are comparable to those of the models with GMamba, and compare their performance with and without GMamba, as shown in Table 22. The results indicate that, although scaling up the baseline models can slightly improve mIoU and mF1, the performance gains brought by GMamba consistently surpass those achieved by merely increasing model capacity. This demonstrates that the advantages of GMamba are not solely due to the increase in parameters, but stem from its enhanced global modeling capability and frequency-guided state space mechanisms, thereby validating its efficiency and general applicability across different backbone architectures.

Table 22: Fair Comparison: Scaled Baselines Matching Params of +GMamba

| Model Variant | Params (M) | mIoU (%) | mF1 (%) |
|---|---|---|---|
| ResNet34 (Baseline) | 25.33 | 81.65 | 89.24 |
| ResNet34 (Scaled, matched params) | 31.06 | 83.18 | 90.59 |
| ResNet34 + GMamba (Ours, matched params) | 30.96 | **84.74** | **91.56** |
| Swin-T (Baseline) | 36.48 | 82.44 | 89.75 |
| Swin-T (Scaled, matched params) | 49.50 | 83.76 | 90.95 |
| Swin-T + GMamba (Ours) | 49.13 | **84.83** | **91.61** |
| ConvNeXt-S (Baseline) | 58.42 | 83.11 | 90.19 |
| ConvNeXt-S (Scaled, matched params) | 71.07 | 84.46 | 91.38 |
| ConvNeXt-S + GMamba (Ours) | 71.06 | **86.00** | **92.31** |

Table 23: Ablation of GMamba with Different Global Modeling Modules

| Global Module | Latency (ms / image) | GPU Memory (GB) | mIoU (%) |
|---|---|---|---|
| Self-Attention (Transformer) | 68.9 | 12.78 | 84.78 |
| VSSM (Mamba) | 51.6 | 8.71 | 84.01 |
| GSSM (GMamba, Ours) | 58.1 | 10.13 | 86.00 |

## K.6 PERFORMANCE ANALYSIS AND EFFICIENCY OF GSSM

To evaluate the efficiency of GSSM, we replaced the GSSM module in GMamba with existing approaches (self-attention and VSSM). Table 23 presents the inference performance and accuracy of the GMamba model with different global modeling modules. All experiments were conducted on the Vaihingen dataset using a single NVIDIA RTX 4090 (24GB) GPU with an input resolution of 1024×1024, and tested on the UNet-ConvNeXt(S) backbone. We measured the average inference latency (ms per image) and peak GPU memory usage (GB) with a batch size of 1. It can be observed that the Transformer-based self-attention module achieves relatively high mIoU (84.78%) but suffers from high latency (68.9 ms per image) and memory consumption (12.78 GB), indicating lower efficiency. The VSSM (Mamba) module exhibits the best inference speed and memory usage (51.6 ms / 8.71 GB), but its accuracy is slightly lower (84.01%). In contrast, our proposed GSSM module achieves a substantial improvement in mIoU (86.00%) while maintaining low latency (58.1 ms) and moderate GPU memory usage (10.13 GB), demonstrating a superior trade-off between efficiency and accuracy. Fig. 13 illustrates the inference performance of GSSM compared to other global modeling modules on a single NVIDIA RTX 4090 (24 GB) GPU across different image resolutions. During the evaluation, the model was set to evaluation mode and gradient computation was disabled. As the resolution increases, both the inference latency and peak GPU memory usage of GSSM exhibit approximately linear growth.

Table 24: Comparison of Frequency Enhancement Strategies in GMamba

| Method | Frequency Strategy | Position | Mechanism | mIoU (%) |
|---|---|---|---|---|
| Baseline (GMamba) | None | None | None | 83.11 |
| GMamba w/ Dual-Branch Fusion | Dual-Branch Fusion | After SSM | Post-hoc feature fusion | 84.10 |
| GMamba w/ Pre-Modulation | Frequency Pre-Modulation | Before SSM | Frequency-guided state update | **86.00** |

## K.7 COMPARISON OF FREQUENCY ENHANCEMENT STRATEGIES

In Table 24, we compare the proposed frequency pre-modulation method with most existing frequency-enhancement strategies, such as dual-branch fusion. The baseline model without any frequency modulation achieves an mIoU of 83.11%. Introducing a dual-branch fusion strategy (post-hoc feature fusion) after the state space model increases the mIoU to 84.10%. In contrast, our approach, which applies frequency pre-modulation prior to SSM to guide state updates, achieves the highest performance, with an mIoU of 86.00%. These results indicate that incorporating frequency information via pre-modulation can more effectively enhance the state modeling process, further validating the effectiveness and rationale of the proposed frequency-modulated SSM framework, which validates our hypothesis.

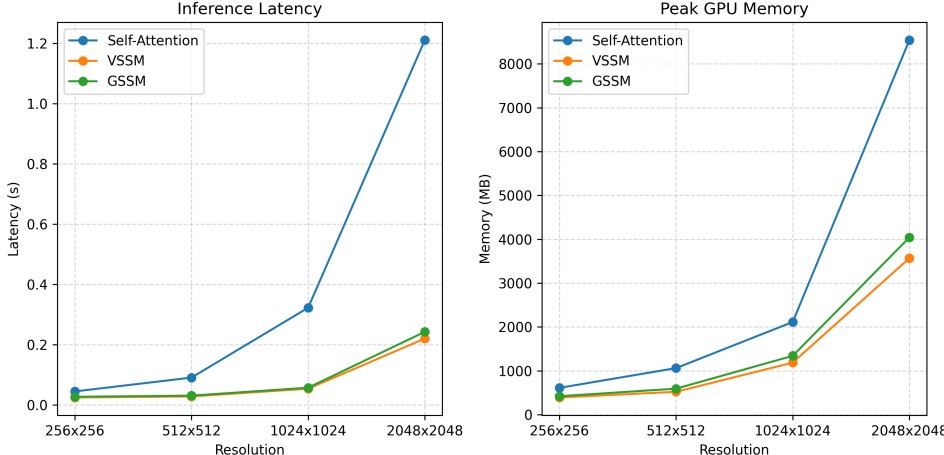

Figure 13: Performance comparison of three global modeling modules under different input resolutions.

## L    FAILURE CASE ANALYSIS

We present some failure cases in Fig. 14. In the first three images, the scenes contain dense crowds. Our model fails to accurately identify individuals in the back rows, whose targets occupy only a few pixels and are partially occluded. In the last image, the scene shows food items on supermarket shelves. These items are small, closely packed, and exhibit similar colors and textures, and were also not accurately recognized. The GSSM module primarily enhances the recursive state selection process under global frequency guidance. For tiny objects, local information is easily diluted through multiple downsampling and state transitions. Consequently, GMamba achieves limited performance improvement in small-object scenarios. Future work could explore integrating more powerful local feature enhancement mechanisms to improve performance on small object recognition.

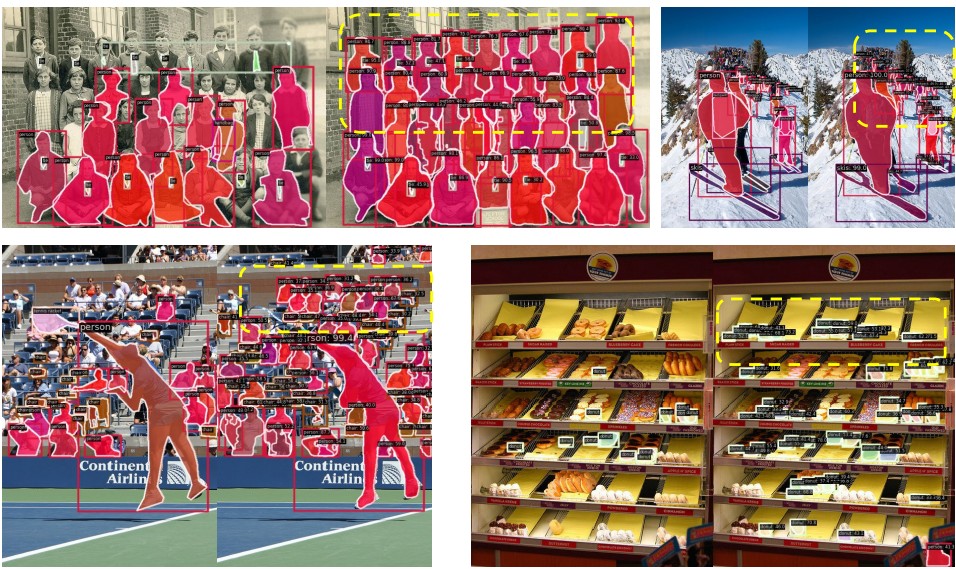

Figure 14: Visualization of failure cases.

