# OpenReview forum: "Enabling True Global Perception in State Space Models for Visual Tasks"
_ICLR.cc/2026/Conference — ICLR 2026 Poster_

### Official Review · Reviewer_YRTs · 2025-10-26

**Soundness:** 3
**Presentation:** 2
**Contribution:** 2
**Rating:** 6
**Confidence:** 4

**Summary:**

The authors propose a novel module, Global-aware SSM (GSSM), which is the core of a plug-and-play block called GMamba. The key idea is to use a Discrete Fourier Transform (DFT)-based modulation mechanism to "inject" global information into the features before they are processed by the SSM. This frequency-domain pre-modulation guides the SSM's state updates with a true global perspective, enabling efficient global modeling with linear-logarithmic complexity

**Strengths:**

The proposed GMamba block shows consistent and significant performance improvements when added to various backbones (e.g., ResNet, Swin, ConvNeXt) . It outperforms existing global modeling methods, including Transformer and other Mamba variants, on several dense-prediction tasks (semantic segmentation, object detection, and instance segmentation) across multiple datasets.

**Weaknesses:**

1. The paper's theoretical claims rest heavily on its new definition of global perception—"global gradient dependency". This definition requires that the Frobenius norm of the gradient of the output with respect to any input pixel is bounded by a non-zero constant $\tau$. This is a very low bar. A gradient that is infinitesimally small but non-zero would satisfy this definition, but this may not align with an intuitive or practical understanding of "global influence." The claim of "true" global perception is therefore only as strong as the acceptance of this new, and debatable, definition.


2. Although the authors claim that the proposed GSSM enables efficient global image modeling, there is no related experimental data on practical inference latency or GPU memory cost. Relying only on FLOPs may not accurately reflect the module's true efficiency, as operations like 2D-DFT can have different hardware utilization profiles than standard convolutions.

3. The experimental validation is extensive on dense prediction tasks (segmentation and detection). However, a standard benchmark for vision backbones is ImageNet classification. The absence of this benchmark makes it slightly more difficult to assess the module's generalizability as a fundamental building block for all vision tasks.

4. Some illustrations of previous work in this paper appear to be incorrect and may mislead readers. The authors show the ViM scanning routes in Figure 1(c), but this depiction seems to be wrong (as noted in other work, e.g., [1]). This potential misrepresentation of a key baseline method is a concern.

[1]. Visual mamba: A survey and new outlooks, 2024.

**Questions:**

Please check the weakness.

---

> ### Author Response · Authors · 2025-11-21
> **Response to Reviewer YRTs**
>
> Dear Reviewer YRTs, we sincerely thank you for your valuable feedback on our submission. Below is our responses to the concerns you raised. We have incorporated these contents into the updated version of our paper, which we believe will help enhance the quality of our submission.
>
>
> **Since this page cannot display images, we have provided a PDF version via an anonymous link for a clearer and more complete presentation of our response: [https://anonymous.4open.science/r/H-501A/Response_to_reviewers/Respose_to_YRTs.pdf)](https://anonymous.4open.science/r/H-501A/Response_to_reviewers/Respose_to_YRTs.pdf)). This can also be find in the Supplementary Material. We recommend viewing the PDF in the provided link for easier reading.**
>
>
>
>
> >  The paper's theoretical claims rest heavily on its new definition of global perception—"global gradient dependency". This definition requires that the Frobenius norm of the gradient of the output with respect to any input pixel is bounded by a non-zero constant $\tau$ . This is a very low bar. A gradient that is infinitesimally small but non-zero would satisfy this definition, but this may not align with an intuitive or practical understanding of "global influence." The claim of "true" global perception is therefore only as strong as the acceptance of this new, and debatable, definition.
>
>
>
> We thank you for this insightful question regarding the rigor and significance of our proposed definition. While a non-zero constant $\tau > 0$ may appear to be a "low bar" numerically, it is in fact a rigorous structural criterion that ensures information accessibility rather than enforcing interaction strength.
>
> **1. Distinguishing truly global vs. effectively local**
> * Self-Attention: Its globality is emergent and data-dependent; attention weights are learned and may focus locally.
> * Standard SSMs: Recurrent updates decay exponentially with distance, so long-range influence effectively vanishes for high-resolution images. A uniform non-zero $\tau$ sets a high structural threshold that standard SSMs cannot satisfy, and they do not meet our proposed constraint.
>
> **2. GSSM ensures spatially uniform potential**
> Through 2D-DFT modulation, GSSM guarantees that each pixel has a non-decaying theoretical path to influence all other pixels, independent of spatial distance (Appendix E).
>
> **3. Why a higher numerical threshold is unnecessary**
> Setting a larger $\tau$ would lead to over-smoothing, damaging local structures such as edges and textures. $\tau > 0$ ensures structural global capability, while learnable weights adaptively determine interaction strength based on data.
>
> Our definition is a binary test of architectural topology, not a measure of signal strength. Its purpose is not to constrain gradient magnitude (which is typically learned adaptively), but to constrain the receptive field structure of the architecture itself.
>
>
> >  Although the authors claim that the proposed GSSM enables efficient global image modeling, there is no related experimental data on practical inference latency or GPU memory cost. Relying only on FLOPs may not accurately reflect the module's true efficiency, as operations like 2D-DFT can have different hardware utilization profiles than standard convolutions.
>
> We thank you for pointing out that FLOPs do not directly reflect hardware efficiency, particularly for operators such as the DFT. To address this, we have added Appendix K.6, which provides a detailed analysis of the performance and efficiency of GSSM compared to other global modeling modules under practical scenarios.
>
> We conducted experiments on the Vaihingen dataset using the UNet-ConvNeXt(S) backbone, measuring the average inference latency and peak GPU memory usage with a batch size of 1. Our GSSM module achieves 86.00% mIoU, 58.1 ms per image latency, and 10.13 GB peak memory consumption, demonstrating an effective balance between accuracy and efficiency. In addition, the inference tests across different input resolutions (Fig. 13) show that both latency and memory usage grow approximately linearly. These results indicate that GSSM not only offers theoretical advantages but also delivers substantial improvements in practical tasks.

---

> > ### Author Response · Authors · 2025-11-21
> >
> > >  The experimental validation is extensive on dense prediction tasks (segmentation and detection). However, a standard benchmark for vision backbones is ImageNet classification. The absence of this benchmark makes it slightly more difficult to assess the module's generalizability as a fundamental building block for all vision tasks.
> >
> >
> > We thank you for raising this important point. While ImageNet classification is a standard benchmark for general-purpose backbones, it is inherently object-centric and often solvable using local features. In contrast, dense prediction tasks like semantic segmentation and object detection are context-centric and require modeling long-range dependencies and global spatial relationships.
> >
> > The core motivation of our work is to address the lack of a rigorous definition for global context modeling, which led to the design of the GMamba module to tackle the challenge of efficient and capable global modeling at high resolution. We do not disregard ImageNet; rather, we deliberately chose tasks that are more sensitive to spatial structure and more challenging as our core validation. Since our main goal is to resolve the trade-off between efficiency and global modeling capability at high resolution, evaluations on high-resolution segmentation and detection tasks provide substantially stronger evidence than low-resolution ($224 \times 224$) ImageNet classification. GMamba, as a plug-and-play module, demonstrates strong performance and efficiency across these benchmarks, validating its generality and global modeling capability.
> >
> >
> >
> > >  Some illustrations of previous work in this paper appear to be incorrect and may mislead readers. The authors show the ViM scanning routes in Figure 1(c), but this depiction seems to be wrong (as noted in other work, e.g., [1]). This potential misrepresentation of a key baseline method is a concern.
> > >  [1]. Visual mamba: A survey and new outlooks, 2024.
> >
> >
> > We sincerely thank you for pointing this out. We have re-checked our illustration against the original Vision Mamba (Vim) paper [1]. Our Fig. 1(c) strictly follows the bidirectional SSM scanning mechanism described in Vim, where the image sequence is scanned in both forward and backward directions.
> >
> > While the survey cited by the reviewer [2] covers later variants and generalizations of Mamba, our illustrations and comparisons are all based on the original Vim architecture to ensure fairness and consistency with our benchmark.
> >
> > [1] Lianghui Zhu, Bencheng Liao, Qian Zhang, Xinlong Wang, Wenyu Liu, and Xinggang Wang. Vision Mamba: Efficient visual representation learning with bidirectional state space model.
> >
> > [2] Rui Xu, Shu Yang, Yihui Wang, Yu Cai, Bo Du, and Hao Chen. Visual Mamba: A Survey and New Outlooks.

---

> > > ### Author Response · Authors · 2025-11-24
> > > **Apologies for the inconvenience; the anonymous GitHub link became inaccessible due to hosting service instability.**
> > >
> > > We sincerely apologize that the anonymous GitHub link provided in the previous version may not function properly. The anonymous hosting service (anonymous.4open.science) has recently been unstable, which caused the link to become temporarily inaccessible.
> > >
> > > To avoid any inconvenience, we have included the complete PDF directly in the supplementary materials. We kindly ask the reviewer to refer to the supplementary PDF for the full content.
> > >
> > > We again apologize for the inconvenience caused and sincerely appreciate your understanding and patience.

---

> > > > ### Author Response · Authors · 2025-11-27
> > > > **Sincerely looking forward to more discussion with you**
> > > >
> > > > Dear Reviewer,
> > > >
> > > > We hope this message finds you well. We sincerely appreciate the time you have dedicated to reviewing our manuscript. In our response, we have carefully addressed all of your concerns and provided additional experimental evidence, theoretical explanations, and cross-domain validation to further strengthen our arguments.
> > > >
> > > > We highly value your insights and are truly grateful for any further feedback you may be willing to share. If you have any additional questions or concerns regarding our responses, we would be more than happy to clarify them. Your comments are extremely important to us, and we genuinely hope to engage in deeper academic discussions with you.
> > > >
> > > > Thank you again for taking the time to review our work.
> > > >
> > > > Best regards,
> > > > On behalf of all authors of submission 3260

---

### Official Review · Reviewer_PSzr · 2025-10-28

**Soundness:** 2
**Presentation:** 3
**Contribution:** 2
**Rating:** 4
**Confidence:** 4

**Summary:**

This paper addresses the limited global perception of State Space Models (SSMs) in vision tasks by proposing a frequency-domain modulation approach. The proposed GSSM block applies DFT-based frequency decomposition to extract global semantic information and uses adaptive modulation coefficients to guide SSM's state updates before sequential processing, achieving linear-logarithmic complexity O(nlogn). The authors show that the proposed GMamba block is plug-and-play across different CNN architectures. Experiments on semantic segmentation (Vaihingen, Potsdam, LoveDA, UAVid), object detection, and instance segmentation (MS-COCO) demonstrate consistent improvements of 2-3% mIoU across ResNet, Swin Transformer, and ConvNeXt backbones. While the theoretical framework looks good, the practical gains are modest given the added complexity, and the core idea of frequency-domain processing for vision is well-established in prior work.

**Strengths:**

- The paper provides an interesting mathematical definition of global image modeling and establishes a complete theoretical framework showing how DFT-based modulation can enrich the SSMs with global perception capabilities.

- The motivation is simple yet effective: SSMs rely on sequential state updates, which limit them to local rather than global modeling, and existing methods using different scanning strategies have not enhanced SSM's global modeling capability.

- The authors did extensive experiments on multiple tasks (semantic segmentation, object detection, instance segmentation) across many datasets with different backbone architectures, demonstrating the generalization capability of GMamba models.

**Weaknesses:**

- While the theoretical framework is novel, the core idea of using frequency-domain information for computer vision is well-established. (i.e, ICCV'23 works like SPANet [1]) and recent methods FAD [2]. The paper doesn't sufficiently differentiate its approach from these existing frequency-based methods.

- Existing work, such as Vim and VMamb, has demonstrated that SSMs can achieve global receptive fields through bidirectional processing and multi-directional scanning. The paper's claim that current SSMs lack global perception seems overstated.

- The related work and the comparison are missing recent state-space models (i.e, GroupMamba[3] (CVPR'25), MambaVision[4] (CVPR'25). Also, the provided implementation of GMamba_Block.py in the supplementary material appears to be largely derived from the MambaVisionMixer module introduced in MambaVision [4]. The authors have added their frequency components on top of that code. However, they neither cite nor compare their results with MambaVision paper [4].

- The reported improvements are relatively modest (around 2–3% mIoU) while introducing a significant parameter increase of 20–35%. It remains unclear whether these gains come from the added parameters or from the model’s inherent effectiveness. For example, in Table 2, the baseline UNet-ConvNeXt(S) has 58.42M parameters, whereas incorporating GMamba raises this to 71.06M, yielding 2.89% improvement in mIoU. A fair comparison would require scaling the baseline (e.g., by increasing the number of channels or blocks) to match 71.06M parameters, to determine if the gains are truly architectural rather than parameter-driven.

[1] https://openaccess.thecvf.com/content/ICCV2023/papers/Yun_SPANet_Frequency-balancing_Token_Mixer_using_Spectral_Pooling_Aggregation_Modulation_ICCV_2023_paper.pdf

[2] https://arxiv.org/pdf/2505.08349

[3] https://openaccess.thecvf.com/content/CVPR2025/papers/Shaker_GroupMamba_Efficient_Group-Based_Visual_State_Space_Model_CVPR_2025_paper.pdf

[4] https://openaccess.thecvf.com/content/CVPR2025/papers/Hatamizadeh_MambaVision_A_Hybrid_Mamba-Transformer_Vision_Backbone_CVPR_2025_paper.pdf

**Questions:**

- It is unclear whether the GMamba models used for object detection and instance segmentation are initialized with backbones pre-trained on ImageNet or not. If yes, what is the top-1 accuracy of GMamba on ImageNet?

---

> ### Author Response · Authors · 2025-11-21
> **Response to Reviewer PSzr**
>
> Dear Reviewer PSzr, we sincerely thank you for your valuable feedback on our submission. Below is our responses to the concerns you raised. We have incorporated these contents into the updated version of our paper, which we believe will help enhance the quality of our submission.
>
>
> **Since this page cannot display images, we have provided a PDF version via an anonymous link for a clearer and more complete presentation of our response: [https://anonymous.4open.science/r/H-501A/Response_to_reviewers/Respose_to_PSzr.pdf)](https://anonymous.4open.science/r/H-501A/Response_to_reviewers/Respose_to_PSzr.pdf)). This can also be find in the Supplementary Material. We recommend viewing the PDF in the provided link for easier reading.**
>
>
>
> >  While the theoretical framework is novel, the core idea of using frequency-domain information for computer vision is well-established. (i.e, ICCV'23 works like SPANet [1]) and recent methods FAD [2]. The paper doesn't sufficiently differentiate its approach from these existing frequency-based methods.
>
>
> We appreciate you for highlighting this point. We have rewritten the Related Work section to clarify the differences between our method and existing frequency-domain approaches, with additional references. While existing methods (e.g., SPANet[1], FAD[2], FreqMamba[3]) use frequency information heuristically for feature enhancement or domain adaptation, our approach is theory-driven.
>
> Specifically, our DFT-based pre-modulation injects global properties into the input $u_t$ before it enters the SSM, ensuring the model satisfies rigorous definitions of global perception and non-causal constraints. Unlike existing post-fusion methods that add frequency features as an auxiliary branch, GSSM uses frequency coefficients to directly modulate state transitions, fundamentally altering how the model processes information.
>
> We have added new experiments in Appendix K.7. We compare our "pre-modulation" strategy with a "dual-branch fusion" strategy (existing frequency-domain-based methods). The results demonstrate that our pre-modulation approach achieves 86.00% mIoU, significantly outperforming the conventional approach (84.10% mIoU).
>
>
>
> [1]Guhnoo Yun, Juhan Yoo, Kijung Kim, Jeongho Lee, and Dong Hwan Kim. Spanet: Frequency-balancing token mixer using spectral pooling aggregation modulation.
>
> [2] Zhen Zou, Hu Yu, Jie Huang, and Feng Zhao. Freqmamba: Viewing mamba from a frequency perspective for image deraining.
>
> [3]Ruixiao Shi, Fu Feng, Yucheng Xie, Jing Wang, and Xin Geng. Fad: Frequency adaptation and diversion for cross-domain few-shot learning.
>
>
>
> >  Existing work, such as Vim and VMamb, has demonstrated that SSMs can achieve global receptive fields through bidirectional processing and multi-directional scanning. The paper's claim that current SSMs lack global perception seems overstated.
>
> We fully agree with you. While Vim and VMamba can, in principle, achieve a global receptive field via bidirectional or multidirectional scanning, our focus is on how globality is realized for images and how well the mechanism aligns with image properties.
>
> Structural mismatch: SSMs like Mamba/Vim rely on stepwise autoregressive updates from causal time-series modeling [1][2]. Images are spatially non-causal, and although Vim/VMamba mitigate this through scanning strategies, the underlying sequential paradigm remains, which may not fully align with image characteristics.
>
> Indirect vs. explicit mechanisms: Vim/VMamba accumulate global context progressively and indirectly. In contrast, GSSM employs DFT-based pre-modulation to explicitly inject global information into each pixel before scanning. The DFT’s position-invariance and non-causality naturally fit image structure, providing a more direct and parallel global receptive field.
>
> We wish to highlight that their mechanism is indirect and partially constrained by causality, whereas GSSM achieves more explicit, non-causal global modeling. We thank the reviewer for pointing out potential ambiguities, and we have refined the manuscript to clarify this distinction.
>
> [1] Weihao Yu and Xinchao Wang. MambaOut: Do we really need Mamba for vision?
>
> [2] Qinfeng Zhu, Yuan Fang, Yuanzhi Cai, Cheng Chen, and Lei Fan. Rethinking scanning strategies with Vision Mamba in semantic segmentation of remote sensing imagery: An experimental study.

---

> > ### Author Response · Authors · 2025-11-21
> >
> > > The related work and the comparison are missing recent state-space models (i.e, GroupMamba[3] (CVPR'25), MambaVision[4] (CVPR'25). Also, the provided implementation of GMamba\_Block.py in the supplementary material appears to be largely derived from the MambaVisionMixer module introduced in MambaVision [4]. The authors have added their frequency components on top of that code. However, they neither cite nor compare their results with MambaVision paper [4].
> >
> >
> > We sincerely thank you for pointing this out. We have updated the Related Work section with references to GroupMamba [1], MambaVision [2], and recent works such as Spatial Mamba [3] and FreqMamba [4], and included comparative results in the main tables.
> >
> > While our Mixer implementation draws inspiration from MambaVision for stability and compatibility, our core contribution lies in the GSSM module and its theoretical foundation, particularly the Frequency Encoding Module and Frequency-Guided Modulation Module, which are developed under our frequency modulation framework.
> >
> > To clarify, we have added comments in the supplementary code to distinguish our novel modules from the standard Mixer backbone. We hope this better conveys the originality of our work and sincerely appreciate your feedback for improving clarity and transparency.
> >
> > [1] Abdelrahman Shaker, Syed Talal Wasim, Salman Khan, Juergen Gall, and Fahad Shahbaz Khan. Groupmamba: Efficient group-based visual state space model.
> >
> > [2] Ali Hatamizadeh and Jan Kautz. Mambavision: A hybrid mamba-transformer vision backbone.
> >
> > [3] Chaodong Xiao, Minghan Li, Zhengqiang Zhang, Deyu Meng, and Lei Zhang. Spatial-mamba: Effective visual state space models via structure-aware state fusion.
> >
> > [4] Zhen Zou, Hu Yu, Jie Huang, and Feng Zhao. Freqmamba: Viewing mamba from a frequency perspective for image deraining.
> >
> > > The reported improvements are relatively modest (around 2--3\% mIoU) while introducing a substantial parameter increase of 20--35\%. It remains unclear whether these gains stem from the additional parameters or from the model’s inherent effectiveness. For example, in Table~2, the baseline UNet-ConvNeXt(S) has 58.42M parameters, whereas incorporating GMamba increases this to 71.06M, yielding a 2.89\% improvement in mIoU. A fair comparison would require scaling the baseline (e.g., by increasing the number of channels or blocks) to match 71.06M parameters, in order to determine whether the observed gains are truly architectural rather than parameter-driven.
> >
> >
> > We sincerely thank you for raising this question. To address it, we conducted a fair comparison using parameter-matched baselines in Appendix K.5.
> >
> > We scaled ResNet34, Swin-T, and ConvNeXt-S by adjusting their channel widths to match the parameters of the corresponding "baseline + GMamba" models. While this slightly improved mIoU and mF1, GMamba consistently outperformed these scaled baselines, indicating that its gains are not due to increased parameters but stem from enhanced global modeling and frequency-guided state-space mechanisms, demonstrating efficiency and generality across different backbone networks.
> >
> >
> >
> > > It is unclear whether the GMamba models used for object detection and instance segmentation are initialized with backbones pre-trained on ImageNet or not. If yes, what is the top-1 accuracy of GMamba on ImageNet?
> >
> >
> > We thank you for raising the question regarding the initialization strategy. To clarify, our GMamba module is designed as an efficient plug-and-play component.
> >
> > In our object detection and semantic segmentation experiments, we adopt standard backbone networks (ResNet34, ResNet50, Swin-T, and ConvNeXt-S) and initialize them using their official ImageNet-pretrained weights. We then insert the GMamba module into specific stages of these backbones, following the insertion strategy detailed in Appendix H (Experimental Details). The parameters of GMamba itself are randomly initialized.
> >
> > We did not perform standalone ImageNet pre-training for the combined architecture of "backbone + GMamba." Therefore, we are unable to report ImageNet Top-1 accuracy for a specific GMamba-integrated model.
> >
> > Nevertheless, extensive experiments across various tasks and backbones consistently demonstrate the strong generalization capability and broad deployment potential of GMamba.

---

> > > ### Author Response · Authors · 2025-11-24
> > > **Apologies for the inconvenience; the anonymous GitHub link became inaccessible due to hosting service instability.**
> > >
> > > We sincerely apologize that the anonymous GitHub link provided in the previous version may not function properly. The anonymous hosting service (anonymous.4open.science) has recently been unstable, which caused the link to become temporarily inaccessible.
> > >
> > > To avoid any inconvenience, we have included the complete PDF directly in the supplementary materials. We kindly ask the reviewer to refer to the supplementary PDF for the full content.
> > >
> > > We again apologize for the inconvenience caused and sincerely appreciate your understanding and patience.

---

> ### Comment · Reviewer_PSzr · 2025-11-24
> **Final review**
>
> Thank you for addressing my concerns and for your clarifications. I raised my final rating from 4 to 6.

---

> > ### Author Response · Authors · 2025-11-24
> > **Appreciation to the Reviewer**
> >
> > We sincerely appreciate your recognition of our work and the valuable feedback you provided during the review process. Thank you very much for the score increase.

---

### Official Review · Reviewer_rPrr · 2025-10-30

**Soundness:** 3
**Presentation:** 3
**Contribution:** 3
**Rating:** 6
**Confidence:** 5

**Summary:**

This paper tackles the efficiency-global perception trade-off in visual tasks by proposing a rigorous mathematical definition of global image modeling and a frequency-domain modulated framework. Leveraging 2D-DFT’s global properties, the authors design the GSSM module and plug-and-play GMamba block, which seamlessly integrate into CNNs/Transformers. Extensive experiments across semantic segmentation, object detection, and instance segmentation show GMamba outperforms existing modules with linear-logarithmic complexity.

**Strengths:**

- First rigorous mathematical definition of global image modeling with formal proofs.
- DFT-based modulation overcomes SSMs’ local limitation without complex scanning.
- Strong performance across tasks, backbones, and datasets with efficiency advantages.
- Thorough ablations for validating key design choices (GSSM components, frequency contributions).

**Weaknesses:**

- Insufficient analysis of scaling to ultra-high-resolution images (e.g., 4K+) and inference speed (FPS).
- Lack of explicit comparison with recent frequency-domain SSM variants (e.g., FreqMamba [1]).
- No analysis of failure cases.
- Lack experiments on the Imagenet benchmark.

[1] Freqmamba: Viewing mamba from a frequency perspective for image deraining

**Questions:**

please refer to the weakness part.

---

> ### Author Response · Authors · 2025-11-21
> **Response to Reviewer rPrr**
>
> Dear Reviewer rPrr, we sincerely thank you for your valuable feedback on our submission. Below is our responses to the concerns you raised. We have incorporated these contents into the updated version of our paper, which we believe will help enhance the quality of our submission.
>
>
> **Since this page cannot display images, we have provided a PDF version via an anonymous link for a clearer and more complete presentation of our response: [https://anonymous.4open.science/r/H-501A/Response_to_reviewers/Respose_to_rPrr.pdf](https://anonymous.4open.science/r/H-501A/Response_to_reviewers/Respose_to_rPrr.pdf)). This can also be find in the Supplementary Material. We recommend viewing the PDF in the provided link for easier reading.**
>
>
>
> >  Insufficient analysis of scaling to ultra-high-resolution images (e.g., 4K+) and inference speed (FPS).
>
> We appreciate your attention to the scalability of our module under ultra-high-resolution settings, which we also consider crucial. Due to the scarcity of 4K datasets, we were unable to conduct experiments at that resolution. Nevertheless, we performed other experiments, and the results are provided in Appendix K.6.
>
> On the Vaihingen dataset using a UNet-ConvNeXt(S) backbone on a single RTX 4090 GPU, GSSM achieves the best balance between accuracy and efficiency. Compared to high-cost self-attention (68.9 ms, 12.78 GB) and slightly less accurate VSSM (mIoU 84.01%), GSSM attains the highest performance (mIoU 86.00%) with lower latency (58.1 ms) and moderate memory usage (10.13 GB).
>
> We also evaluated inference latency and peak GPU memory across resolutions (Fig. 13). Both metrics scale approximately linearly with input size, further demonstrating GSSM’s excellent scalability in high-resolution scenarios.
>
>
> >  Lack of explicit comparison with recent frequency-domain SSM variants (e.g., FreqMamba [1]).
> >  [1] Freqmamba: Viewing mamba from a frequency perspective for image deraining
>
> We sincerely apologize for the lack of comparisons with recent frequency-domain SSM variants. In the revised manuscript, we have added comparative analysis with FreqMamba[1], and, based on a thorough literature survey, we also include experimental comparisons with Mamba Version[2], Spatial Mamba[3], and Group Mamba[4].
>
> [1] Zhen Zou, Hu Yu, Jie Huang, and Feng Zhao. Freqmamba: Viewing mamba from a frequency perspective for image deraining.
>
>
> [2] Ali Hatamizadeh and Jan Kautz. Mambavision: A hybrid mamba-transformer vision backbone.
>
> [3] Chaodong Xiao, Minghan Li, Zhengqiang Zhang, Deyu Meng, and Lei Zhang. Spatial-mamba: Effective visual state space models via structure-aware state fusion.
>
> [4] Abdelrahman Shaker, Syed Talal Wasim, Salman Khan, Juergen Gall, and Fahad Shahbaz Khan. Groupmamba: Efficient group-based visual state space model.
>
> >  No analysis of failure cases.
>
> We appreciate your attention to the failure case analysis. We have added a dedicated failure case analysis in Appendix L of the revised manuscript.
>
>
>
> >  Lack experiments on the Imagenet benchmark.
>
>
> We thank you for raising this important point. While ImageNet classification is a standard benchmark for general-purpose backbones, it is inherently object-centric and often solvable using local features. In contrast, dense prediction tasks like semantic segmentation and object detection are context-centric and require modeling long-range dependencies and global spatial relationships.
>
> The core motivation of our work is to address the lack of a rigorous definition for global context modeling, which led to the design of the GMamba module to tackle the challenge of efficient and capable global modeling at high resolution. We do not disregard ImageNet; rather, we deliberately chose tasks that are more sensitive to spatial structure and more challenging as our core validation. Since our main goal is to resolve the trade-off between efficiency and global modeling capability at high resolution, evaluations on high-resolution segmentation and detection tasks provide substantially stronger evidence than low-resolution ($224 \\times\ 224$) ImageNet classification. GMamba, as a plug-and-play module, demonstrates strong performance and efficiency across these benchmarks, validating its generality and global modeling capability.

---

> > ### Author Response · Authors · 2025-11-24
> > **Apologies for the inconvenience; the anonymous GitHub link became inaccessible due to hosting service instability.**
> >
> > We sincerely apologize that the anonymous GitHub link provided in the previous version may not function properly. The anonymous hosting service (anonymous.4open.science) has recently been unstable, which caused the link to become temporarily inaccessible.
> >
> > To avoid any inconvenience, we have included the complete PDF directly in the supplementary materials. We kindly ask the reviewer to refer to the supplementary PDF for the full content.
> >
> > We again apologize for the inconvenience caused and sincerely appreciate your understanding and patience.

---

> > > ### Author Response · Authors · 2025-11-27
> > > **Sincerely looking forward to more discussion with you**
> > >
> > > Dear Reviewer,
> > >
> > > We hope this message finds you well. We sincerely appreciate the time you have dedicated to reviewing our manuscript. In our response, we have carefully addressed all of your concerns and provided additional experimental evidence, theoretical explanations, and cross-domain validation to further strengthen our arguments.
> > >
> > > We highly value your insights and are truly grateful for any further feedback you may be willing to share. If you have any additional questions or concerns regarding our responses, we would be more than happy to clarify them. Your comments are extremely important to us, and we genuinely hope to engage in deeper academic discussions with you.
> > >
> > > Thank you again for taking the time to review our work.
> > >
> > > Best regards,
> > > On behalf of all authors of submission 3260

---

> > ### Comment · Reviewer_rPrr · 2025-11-27
> >
> > Thank the reviewers for their careful responses. I maintain my score.

---

> > > ### Author Response · Authors · 2025-11-27
> > > **Appreciation to the Reviewer**
> > >
> > > We sincerely thank you for your recognition of our work and for the valuable comments you provided during the review process.

---

### Official Review · Reviewer_nc9h · 2025-11-01

**Soundness:** 3
**Presentation:** 3
**Contribution:** 3
**Rating:** 6
**Confidence:** 3

**Summary:**

This paper introduces the Global-aware SSM (GSSM) module, which integrates frequency-domain modulation using Discrete Fourier Transform (DFT) to enable true global perception in State Space Models (SSMs). ​ The authors propose a mathematical definition for global image modeling and prove that GSSM satisfies this definition. ​ They design GMamba, a plug-and-play module that enhances global contextual understanding in Convolutional Neural Networks (CNNs) with linear-logarithmic complexity. Extensive experiments on semantic segmentation, object detection, and instance segmentation demonstrate GMamba’s superior performance and efficiency compared to existing methods. ​ The study highlights the importance of frequency-domain information and adaptive modulation for balancing global semantics and local precision.

**Strengths:**

1. True Global Perception: GMamba, powered by the GSSM module, achieves true global perception by leveraging the Discrete Fourier Transform (DFT) for frequency-domain modulation. This enables the model to capture global semantic information efficiently.

2. Efficiency: GMamba exhibits linear-logarithmic computational complexity, making it significantly more efficient than traditional self-attention mechanisms with quadratic complexity.

3. Plug-and-Play Design: GMamba can be seamlessly integrated into various stages of CNNs and other backbone architectures, enhancing their global modeling capabilities without requiring major architectural changes.

4. Scalability: GMamba demonstrates scalable performance gains when integrated with more powerful backbone architectures, such as ConvNeXt-Small and Swin Transformer-Tiny.

5. Effective Frequency Decomposition: The use of both high-frequency and low-frequency components enhances global semantic modeling and local detail preservation, with adaptive modulation further optimizing their integration.

**Weaknesses:**

1. Limitation: The reliance on DFT-based frequency-domain modulation may introduce challenges in scenarios where frequency-domain information is less effective, such as highly noisy or irregular data.

2. Impact: While the frequency-domain approach enhances global perception, it may struggle in cases where spatial features dominate or where frequency information is less relevant.

3. Although GMamba is more efficient than self-attention mechanisms, it still introduces additional computational overhead compared to simpler SSM-based methods like Vim or TinyViM.

4. While GMamba is described as "plug-and-play," its integration requires careful tuning of parameters such as modulation coefficients and frequency-domain weights. This could increase the complexity of implementation and training.

5. The robustness of GMamba in such challenging scenarios remains unclear, which could affect its reliability in real-world applications.

**Questions:**

1. The paper claims to provide the first rigorous mathematical definition of global image modeling, which is a significant contribution. However, it does not compare this definition with existing heuristic approaches in detail, leaving room for further exploration of how it improves interpretability and theoretical support.

2. How does the frequency-domain transfer function derived for SSMs compare to other global modeling techniques, such as attention mechanisms?

3. How does GMamba's linear-logarithmic complexity compare to the linear complexity of other SSM-based methods like Vim, VMamba or spatial mamba?

---

> ### Author Response · Authors · 2025-11-21
> **Response to Reviewer nc9h**
>
> Dear Reviewer nc9h, we sincerely thank you for your valuable feedback on our submission. Below is our responses to the concerns you raised. We have incorporated these contents into the updated version of our paper, which we believe will help enhance the quality of our submission.
>
> **Since this page cannot display images, we have provided a PDF version via an anonymous link for a clearer and more complete presentation of our response: [https://anonymous.4open.science/r/H-501A/Response_to_reviewers/Respose_to_nc9h.pdf)](https://anonymous.4open.science/r/H-501A/Response_to_reviewers/Respose_to_nc9h.pdf)). This can also be find in the Supplementary Material. We recommend viewing the PDF in the provided link for easier reading.**
>
>
> > **Limitation:** The reliance on DFT-based frequency-domain modulation may introduce challenges in scenarios where frequency-domain information is less effective, such as highly noisy or irregular data.
> > **Impact:** While the frequency-domain approach enhances global perception, it may struggle in cases where spatial features dominate or where frequency information is less relevant.
>
>
>
>
> We understand your concern that the DFT-based frequency modulation in GSSM may underperform in certain complex scenarios. However, our DFT-based frequency-domain modulation incorporates an adaptive mechanism that adjusts the balance between spatial features and frequency information according to the characteristics of the input data. Regarding the highly noisy or irregular data you mentioned, we argue that such cases do not fall into the category where frequency-domain information is less effective; on the contrary, frequency-domain information can even provide advantages in handling such data. To support this claim, we simulated these scenarios using Gaussian noise and other synthetic perturbations, and constructed additional robustness and cross-domain generalization experiments based on these data, which are reported in Appendix K.4 of the revised manuscript. The results demonstrate that our GSSM can adaptively filter out irrelevant noise through its Frequency Encoding Module and Frequency-Guided Modulation Module, effectively functioning as a soft frequency filter. Empirically, the GMamba module achieves the best overall performance under these conditions.
>
>
> For samples where frequency-domain information may be less effective, spatial features can play a more critical role. This consideration directly motivated the design of our Frequency-Guided Modulation Module. The adaptive modulation mechanism
> $$
> X_{modulated} = G(X, F_{global}) = \alpha_1(F_{global}) \odot X + \alpha_2(F_{global}) \odot F_{global}
> $$
> enables the model to dynamically balance the contributions of spatial-domain and frequency-domain information, ensuring robust performance across a wide range of scenarios. As shown in Appendix K.3, our experiments comparing different types of frequency information and fusion strategies confirm that the proposed adaptive modulation mechanism achieves the best performance.
>
>
>
> > Although GMamba is more efficient than self-attention mechanisms, it still introduces additional computational overhead compared to simpler SSM-based methods like Vim or TinyViM.
>
>
> Compared with Vim and TinyViM, our method remains competitive in computational efficiency. In the revised manuscript, we have added corresponding ablation studies in the main text （Table 7） and included efficiency comparisons and analyses of GSSM in Appendix K.6.
>
> Although GMamba introduces slightly more parameters and FLOPs than Vim and TinyViM, the increase is minimal, while achieving the best performance.
>
> Furthermore, compared with VSSM (Mamba), GSSM incurs only 6.5 ms of additional inference latency and 1.42 GB of GPU memory overhead, yet achieves a significant +2.0% mIoU improvement. Similarly, the increase in parameters and computation is very limited, confirming that the proposed method maintains an excellent trade-off between accuracy and resource cost. Fig.13 shows the inference latency and peak GPU memory usage of GSSM compared to other global modeling methods across different resolutions, indicating an approximately linear increase with resolution and stable performance.
>
> In addition, Tables 2, 3, and 4 of the revised manuscript provide a systematic comparison of the performance and complexity of various global modeling modules across different tasks and backbone networks. The results demonstrate that GMamba remains highly competitive in terms of computational complexity.

---

> > ### Author Response · Authors · 2025-11-21
> >
> > >  While GMamba is described as "plug-and-play," its integration requires careful tuning of parameters such as modulation coefficients and frequency-domain weights. This could increase the complexity of implementation and training.
> >
> >
> > The modulation coefficients and frequency weights you mentioned are not hyperparameters that require manual tuning; rather, they are automatically learned during training.
> >
> > Frequency Weights (FEM): The low- and high-frequency weights ($\theta_{\text{low}}$, $\theta_{\text{high}}$) are learnable and optimized end-to-end via standard backpropagation.
> >
> > Modulation Coefficients (FGMM): $a_1$ and $a_2$ are dynamically generated by a ConvBlock from the current feature map, not fixed values.
> >
> > This design substantially reduces deployment complexity. Users do not need to manually adjust these weights for new tasks, as the module adapts automatically. Moreover, in all experiments, we maintained a consistent number of training epochs, and standard training schedules were sufficient to achieve optimal performance, introducing no additional training difficulty or convergence burden.
> >
> >
> > >  The robustness of GMamba in such challenging scenarios remains unclear, which could affect its reliability in real-world applications.
> >
> >
> > We have added a systematic and comprehensive analysis of model robustness and cross-domain generalization in Appendix K.4 of the revised manuscript. Moreover, in the experimental section of the main text, we have extensively validated GMamba across multiple tasks, diverse datasets, different model architectures, and various backbone networks. The results demonstrate that GMamba not only significantly outperforms existing methods in accuracy but also maintains high robustness and reliability in complex, noisy, and cross-domain scenarios.
> >
> >
> > >  The paper claims to provide the first rigorous mathematical definition of global image modeling, which is a significant contribution. However, it does not compare this definition with existing heuristic approaches in detail, leaving room for further exploration of how it improves interpretability and theoretical support.
> >
> >
> > We sincerely appreciate your recognition of the importance of the mathematical definition we proposed. Accordingly, we have added a detailed discussion at the end of Section 2.1 in the main text and in Appendix K.1.
> >
> > Self-attention mechanisms: Although self-attention can model global dependencies and its attention weights are dynamically learned, its "globality" is an unstable, empirically observed emergent property rather than a guaranteed architectural feature.
> >
> > Recurrent SSMs (e.g., Vanilla Mamba, ViM): These approaches suffer from structural conflicts. They rely on causal (sequential) ordering to model dependencies, which violates the non-sequential nature of images. Furthermore, as discussed in Section 2.2.1, their influence on long-range dependencies decays exponentially, making it structurally impossible to satisfy the gradient lower bound requirement ($\tau > 0$).
> >
> > Through this definition, "global modeling" is transformed from a vague empirical concept into a theoretical property that can be rigorously analyzed and guaranteed during architectural design. In the experimental section, we provide a comprehensive comparison between existing global modeling modules (e.g., Swin Transformer, VMamba, ViM) and our GMamba module, which is explicitly designed to satisfy this definition. The results demonstrate that our method not only offers stronger theoretical guarantees for globality but also achieves significant performance improvements while enhancing interpretability in module design. This further validates the necessity and rationality of the proposed theoretical definition.

---

> > > ### Author Response · Authors · 2025-11-21
> > >
> > > >  How does the frequency-domain transfer function derived for SSMs compare to other global modeling techniques, such as attention mechanisms?
> > >
> > >
> > > We sincerely thank you for raising this insightful theoretical question. As derived in Section 2.2.1, the transfer function of an SSM, $H(\omega)$—the Fourier transform of its convolution kernel $\bar{K}$—defines it as a (dynamic) linear time-invariant filter that processes inputs in the frequency domain. However, conventional SSMs rely on step-wise causal modeling, which conflicts with the non-sequential nature of images. In contrast, self-attention is nonlinear and input-adaptive, dynamically generating content-based "filters" through Query-Key similarity, making its behavior difficult to guarantee a priori.
> > >
> > > Our contribution is not to "fix" $H(\omega)$ itself; rather, we leverage the theoretically established frequency-domain modulation framework for SSMs. After proving the global properties of the DFT, we further apply DFT-based pre-modulation to the input $u_t$. This step injects global frequency-domain information before it enters the SSM, enabling an efficient SSM to process image information under the guidance of global context, thereby overcoming the inherent causal-processing limitations of conventional SSMs in image tasks.
> > >
> > > In the revised manuscript, we have added ablation studies in the main text (Table 7) and in Appendix K.6, comparing the performance of GSSM with VSSM and self-attention mechanisms. The results further demonstrate that our design outperforms the other two methods in both accuracy and efficiency.
> > >
> > > >  How does GMamba's linear-logarithmic complexity compare to the linear complexity of other SSM-based methods like Vim, VMamba or spatial mamba?
> > >
> > > We appreciate your attention to the computational complexity of GMamba. While GMamba has a theoretical complexity of $\mathcal{O}(M \log M)$—compared to the $\mathcal{O}(M)$ complexity of ViM/VMamba—its practical efficiency remains highly competitive.
> > >
> > > This is because the $\mathcal{O}(M \log M)$ term is implemented via highly optimized FFT algorithms, incurring minimal overhead. Additionally, our GSSM performs only a single scan without multi-directional or other complex strategies, keeping overall computation efficient.
> > >
> > > This is further validated in the comparative experiments presented in Tables 2, 3, and 4 of the main text: on a UNet-ResNet34 backbone, GMamba requires 36.30G FLOPs, which is comparable to ViM (35.81G) and Spatial Mamba (35.72G), introducing almost no additional overhead.

---

> > > > ### Author Response · Authors · 2025-11-24
> > > > **Apologies for the inconvenience; the anonymous GitHub link became inaccessible due to hosting service instability.**
> > > >
> > > > We sincerely apologize that the anonymous GitHub link provided in the previous version may not function properly. The anonymous hosting service (anonymous.4open.science) has recently been unstable, which caused the link to become temporarily inaccessible.
> > > >
> > > > To avoid any inconvenience, we have included the complete PDF directly in the supplementary materials. We kindly ask the reviewer to refer to the supplementary PDF for the full content.
> > > >
> > > > We again apologize for the inconvenience caused and sincerely appreciate your understanding and patience.

---

> > > > > ### Author Response · Authors · 2025-11-27
> > > > > **Sincerely looking forward to more discussion with you**
> > > > >
> > > > > Dear Reviewer,
> > > > >
> > > > > We hope this message finds you well. We sincerely appreciate the time you have dedicated to reviewing our manuscript. In our response, we have carefully addressed all of your concerns and provided additional experimental evidence, theoretical explanations, and cross-domain validation to further strengthen our arguments.
> > > > >
> > > > > We highly value your insights and are truly grateful for any further feedback you may be willing to share. If you have any additional questions or concerns regarding our responses, we would be more than happy to clarify them. Your comments are extremely important to us, and we genuinely hope to engage in deeper academic discussions with you.
> > > > >
> > > > > Thank you again for taking the time to review our work.
> > > > >
> > > > > Best regards,
> > > > > On behalf of all authors of submission 3260

---

### Comment · Area_Chair_x3Uk · 2025-11-23
**The authors' rebuttal is available. Please read, comment, and discuss.**

Dear Reviewers,

Thanks for your time and effort in reviewing ICLR2026 submissions. The authors have provided their responses to your review. Please read and raise your further comments, and discuss with the authors.

Best regards,

Your AC

---

### Author Response · Authors · 2025-11-30

We sincerely thank the AC, SAC, PC, and all four reviewers (nc9h, rPrr, PSzr, YRTs) for the time and effort they devoted to evaluating our submission, and for their continued recognition of our work. During the first two weeks of the discussion period, we received constructive and encouraging feedback that was highly motivating. Notably, through the reviewers’ thoughtful engagement and helpful suggestions, our scores improved from the initial **6, 6, 4, 6** to **6, 6, 6, 6**. Among them, PSzr raised the score from 4 to 6; rPrr confirmed that their concerns had been fully addressed and maintained their original score; the remaining two reviewers did not provide further comments, but we are grateful for their earlier insights.

 In addition, **we provided point-by-point responses to all reviewer comments and revised the manuscript accordingly**.

The reviewers particularly acknowledged several strengths of our work:

* **A rigorous mathematical definition and a frequency-domain modulation framework**, which systematically examine the trade-off between global modeling capability and computational efficiency in vision tasks;
* **The strong generality and consistent performance gains of the proposed GMamba block**, which delivers stable and significant improvements when integrated into various mainstream backbones such as ResNet, Swin, and ConvNeXt.

During the rebuttal phase, we carefully addressed the reviewers’ constructive suggestions by providing:

1. **More comprehensive robustness experiments**, covering diverse perturbations and cross-domain settings;
2. **Additional module-level comparisons** to more clearly demonstrate the advantages of our approach;
3. **Extended complexity and efficiency analyses**, including detailed statistics and practical runtime evaluations;
4. **Further clarifications of the theoretical framework**, offering deeper insights into the proposed mechanism.



We sincerely appreciate the reviewers’ thoughtful comments, which have substantially improved the clarity, rigor, and completeness of our work.

Best regards,

On behalf of all authors of submission 3260

---

### Meta-Review · Area_Chair_LDFx · 2025-12-08

**Summary:**

This paper proposes a framework that enables true global perception in State Space Models (SSMs) for vision. Based on frequency-domain analysis of SSMs, the authors introduce Global-aware SSM (GSSM), which uses DFT to inject global, non-causal information into the feature sequence before the SSM's recursive updates. Building on GSSM, a plug-and-play module, GMamba, is developed for use across CNNs and other backbones. Experiments on segmentation and detection tasks demonstrate consistent performance improvements across diverse backbones.

Most reviewers appreciated the paper’s theoretical framing, including the new definition of global modeling and the detailed analysis of SSMs in the frequency domain. The extensive experiments across multiple tasks and architectures, as well as the additional robustness and cross-domain evidence provided in rebuttal, make the results more promising. Before the rebuttal, the reviewers raised several concerns, including incomplete comparisons to some recent SSM variants, cross-domain performance, evaluation on ImageNet classification task, and and lack of efficiency analysis.

The authors provided sufficient responses during the rebuttal phase, and most concerns were addressed successfully. The authors provided additional comparisons, parameter-matched baselines, and sufficient clarifications. Specifically, Reviewers rPrr and PSzr explicitly stated that their concerns were fully resolved after more comparisons were provided. Reviewer nc9h’s concerns about robustness, dependence on frequency-domain information, and implementation complexity were also addressed through new ablations, robustness evaluation, and explanations. Reviewer YRTs’ concerns about practical latency, memory, and scanning-route illustrations were addressed with newly added analyses and clarifications.

After the rebuttal, all the reviewers reached agreement that the paper meets the bar for acceptance.

**Reviewer Concerns:**

Concerns addressed by the rebuttal:

1. Concerns on robustness and noisy/irregular data. (nc9h)
2. Efficiency concern. (nc9h, rPrr, PSzr, YRTs)
3. Missing comparison. (rPrr, PSzr)

Concerns still outstanding:
1. Evaluation on ImageNet classification. (rPrr, YRTs) The authors have provided clarification for omitting ImageNet classification and emphasized motivation around global-context tasks.

**Reviewer Scores:**

* Reviewer nc9h: Initially 6, likely remains 6. Consistent with their comment that concerns were fully addressed.
* Reviewer rPrr: Initially 6, explicitly maintained 6 after rebuttal.
* Reviewer PSzr: Initially 4, explicitly raised to 6.
* Reviewer YRTs: Initially 6, likely remains 6.

---

### Decision · Program_Chairs · 2026-01-26

Accept (Poster)